# Modelling 3D saccade generation by feedforward optimal control

**Akhil John**[1], **Carlos Aleluia**[1], **A. John Van Opstal**[2], **Alexandre Bernardino**[1]*

**1** Institute for Systems and Robotics, Instituto Superior Técnico, Lisboa. Portugal, **2** Department of Biophysics, Donders Centre for Neuroscience, Radboud University, Nijmegen, The Netherlands

* alex@isr.tecnico.ulisboa.pt

**Data Availability Statement:** The data underlying the results presented in the study are available from: https://bitbucket.org/alexbernardino/saccade3dplos/.

## Abstract

An interesting problem for the human saccadic eye-movement system is how to deal with the degrees-of-freedom problem: the six extra-ocular muscles provide three rotational degrees of freedom, while only two are needed to point gaze at any direction. Measurements show that 3D eye orientations during head-fixed saccades in far-viewing conditions lie in Listing's plane (LP), in which the eye's cyclotorsion is zero (Listing's law, LL). Moreover, while saccades are executed as single-axis rotations around a stable eye-angular velocity axis, they follow straight trajectories in LP. Another distinctive saccade property is their nonlinear main-sequence dynamics: the affine relationship between saccade size and movement duration, and the saturation of peak velocity with amplitude. To explain all these properties, we developed a computational model, based on a simplified and upscaled robotic prototype of an eye with 3 degrees of freedom, driven by three independent motor commands, coupled to three antagonistic elastic muscle pairs. As the robotic prototype was not intended to faithfully mimic the detailed biomechanics of the human eye, we did not impose specific prior mechanical constraints on the ocular plant that could, by themselves, generate Listing's law and the main-sequence. Instead, our goal was to study how these properties can emerge from the application of optimal control principles to simplified eye models. We performed a numerical linearization of the nonlinear system dynamics around the origin using system identification techniques, and developed open-loop controllers for 3D saccade generation. Applying optimal control to the simulated model, could reproduce both Listing's law and and the main-sequence. We verified the contribution of different terms in the cost optimization functional to realistic 3D saccade behavior, and identified four essential terms: total energy expenditure by the motors, movement duration, gaze accuracy, and the total static force exerted by the muscles during fixation. Our findings suggest that Listing's law, as well as the saccade dynamics and their trajectories, may all emerge from the same common mechanism that aims to optimize speed-accuracy trade-off for saccades, while minimizing the total muscle force during eccentric fixation.

**Funding:** A. John Van Opstal is the recipient of an ERC (European Research Council - https://erc. europa.eu) Advanced Grant (693400 ORIENT) of the H2020 Programme. The funders had no role in study design, data collection and analysis, decision to publish, or preparation of the manuscript.

**Competing interests:** The authors have declared that no competing interests exist.

## Author summary

Saccades are rapid eye movements that humans and other animals perform three to four times per second to scan and perceive the environment around them. These movements orient the eye in space with high precision and in a highly stereotyped fashion. Existing studies on animal models advocate that both mechanical and neuronal functions play an important role in the control of the saccades, but some facts are still not fully understood due to difficulties in experimenting and measuring the variables in living animals. Instead, robots are computational and physical models of reality that expose all its variables and can be programmed in interpretable ways. We have built a robotic model of an artificial eye containing the basic ingredients of human eyes: full 3D rotations, viscous friction and 6 muscle-like actuators connected to the eyeball in a geometry similar to the biological system. By synthesizing robotic eye control systems we found that important characteristics of the movements become similar to human saccades when the control relies on few simple fundamental principles: the maximization of saccade accuracy and the minimization of saccade duration, energy in control, and force in the muscles during fixation.

## Introduction

This paper presents a computational model for the control of gaze in a simplified and scaled 3D mechanical eye model with tendon-driven actuation that approximates the extra-ocular muscle geometry of the human eye (see Figs 1 and 2). We propose an optimal open-loop control scheme that, when appropriately configured with cost terms that consider the minimization of accuracy, energy, saccade duration, and force in the muscles, generates the main kinematic and dynamics properties of saccadic eye movements found in primates. Previous studies either have focused on simpler 1D models (vertical/horizontal saccades) or analysed the kinematics and dynamic of saccade behavior independently. Our proposal demonstrates the explanatory power of unifying optimal control principles in both the kinematics and dynamics of a mechanical eye model that executes 3D saccade trajectories.

## Saccade control

In order to generate a goal-directed movement, the brain has to resort to sophisticated neural control strategies that take into account the excess number of degrees of freedom, the non-commutativity of rotational kinematics, and the complex dynamics of the end effectors. The human oculomotor system is a well-studied example. The human eye has six extra-ocular muscles that can each pull the eye in a different direction (Fig 1). Because the eye is fully encased within its bony orbit, it cannot translate [1], thereby reducing the six degrees of freedom for rigid-body motion to three rotational degrees of freedom. However, to direct the gaze-line at any direction, only two coordinates need to be specified, e.g., the azimuth ($\theta$) and elevation ($\varphi$) angles, leaving the ocular rotation around the line of sight (cyclo-torsion, $\psi$) unspecified. If not controlled, however, this poses a potential problem for eye movements, as mathematics holds that sequences of rigid-body rotations in general do not commute. Thus, the order in which the rotations are executed determines the final body orientation. In principle, this could lead to an unwanted accumulation of ocular torsion and significant localization errors [2, 3]. It was already recognized by Donders and Helmholtz that the eye restricts its torsional roll angle in a unique way, depending only on the gaze direction, regardless the path taken by the eye to

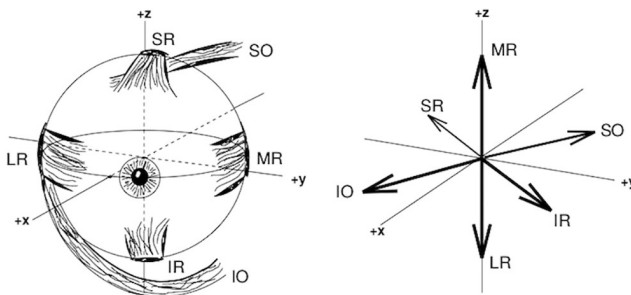

**Fig 1.** Left: Frontal view of the extra-ocular muscles of the right human eye, rotated by about 20 deg leftward from the primary (+*x*) direction—MR (Medial Rectus), LR (Lateral Rectus), SR (Superior Rectus), IR (Inferior Rectus), SO (Superior Oblique), IO (Inferior Oblique). Right: Schematic of the muscle rotation axes, when starting from the primary direction. Right-hand (*x*, *y*, *z*) convention. The ML/LR muscles mainly rotate the eye horizontally (*Z*-direction). The IR/SR, and the IO/SO muscle pairs induce both vertical (*y*) and cyclotorsional (*x*) rotation components. The superior and inferior obliques (SO/IO) divert their pulling trajectories medially: the IO is attached to the medial side of the orbit, while the SO is diverted through a medial pulley, called the trochlea.

reach that orientation (e.g. [4–7]). This observation has been known as Donders' law: $\psi = f(\theta, \varphi)$. Thus, somehow the eye takes care of the noncommutativity problem.

The saccade control problem has also been studied in Robotics. Gaze has been found to play an important role in social interaction, for instance in signalling attitudes, affects, and emotions, and in complementing speech with emphasis and synchronization [8]. This fact has

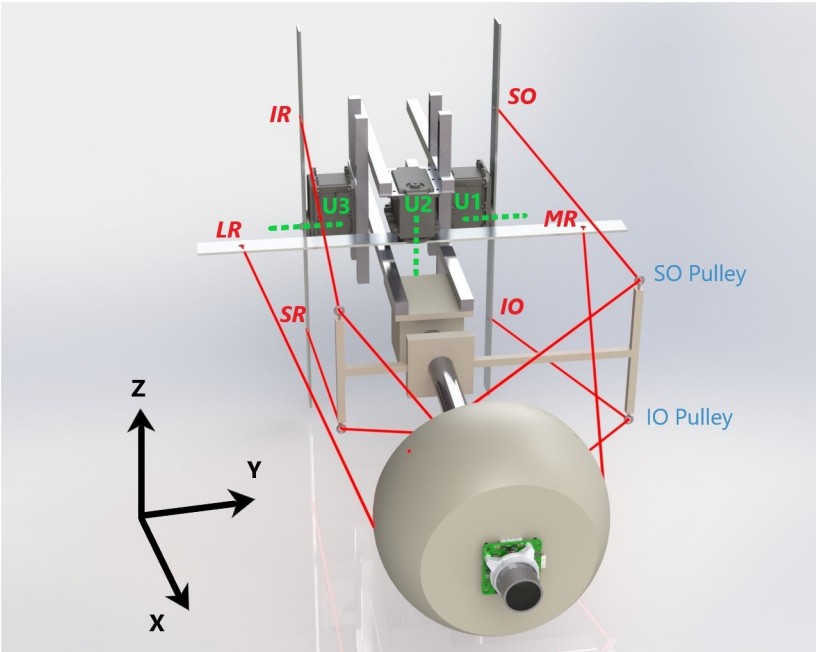

**Fig 2. Mechanical robotic system that serves as a biomimetic prototype for the human eye in Fig 1.** The scaled eye represented by the spherical segment (7 cm radius) with a camera in the front is driven by elastic tendons (in red) that correspond to the six extra-ocular muscles shown in Fig 1. Each pair of muscles are connected to attachment points on metal rods fixed to the motors(U1-U3), with the rotation axes shown by the green dotted lines. The SO,IO,IR and SR muscles pass through fixed pulleys between the attachment points on the eye and the motor rods to adjust their pulling directions.

spawned numerous studies of gaze behaviour during human-robot interaction (see [9], for a review; [10]), not only for letting robots understand human gaze signals, but also (and more important for the present paper) to allow robots to display legible gaze behavior for a human interlocutor. One line of research tries to generate human-like behaviors in robots by recording human motions in the execution of particular tasks [11–13]. Another line of research tries to replicate the observed properties of human gaze behavior by directly programming the theoretical control laws into the robots [14, 15].

While learning the gaze kinematics from humans, or directly programming the theoretical neuroscientific findings in robots are both viable approaches for designing working systems, they are limited in two aspects. First, the underlying principles that rule their emergence are left uncovered. Second, existing models address only very simplified settings.

The present paper considers the generation of saccades with fully unconstrained kinematics and develops a methodology to generate control systems that both satisfy mechanical optimality criteria (energy and stress), and performance criteria (duration and accuracy), while mimicking the kinematics and transient properties observed in natural gaze behavior. To validate our approach we have built a scaled up robotic prototype driven by three motors attached to elastic tendons, schematically shown in Fig 2. The tendons are attached to an artificial eye ball in a way that, although not matching its biological counterpart, shares a similar geometry. The other end of the tendons are connected in pairs to symmetric points on rods that are fixed to each motor. Unlike the biological muscles, these elastic tendons can only apply force when stretched so they are actuated in pairs by the rotation of the motor rods. The tendons for vertical and torsional muscles pass through fixed routing points to adjust their pulling directions to provide enough ocular range. We built a simulation model of the system in order to study the role of optimal control methods in the emergence of relevant oculomotor behaviors.

The proposed approach may lead to more energy efficient and more durable robots, with more flexibility in replicating the complex repertoire of oculomotor behaviors exhibited by humans. Building humanoid-representative oculomotor robots may also be useful for ground-truth benchmarking of many eye tracking systems.

## 3D kinematics

As will be described in more detail below, a 3D eye orientation is most efficiently described by a single-axis rotation of the eye from the primary position. Thus, eye orientations can be parametrized by the direction of the rotation axis, $\hat{\mathbf{n}}$, $\|\hat{\mathbf{n}}\| = 1$, and the amount of rotation, $\rho$, around that axis needed to attain the orientation: $f(\hat{\mathbf{n}}, \rho) = \rho\hat{\mathbf{n}} = \mathbf{e} = (e_x, e_y, e_z)$, the so-called Euler vector, with $(x, y, z)$ a head-fixed, right-handed Cartesian reference frame ($x$ = torsion, $y$ = vertical, $z$ = horizontal rotation). Due to a closer relationship to quaternions, it is convenient to adopt the equivalent half-radian rotation vector representation [16]:

$\mathbf{r} = (r_x, r_y, r_z) = \tan(\rho/2)\hat{\mathbf{n}}$.

With the head upright and not moving, and gaze directed at infinity, Donders' law is further constrained by stating that the rotation axes describing all eye orientations are restricted to a plane (Listing's law (LL); [4–6, 17]). Fig 3 shows an example of more than 3500 rotation axes, computed during fixations and saccades made by a monkey in the lab (head-restrained, spontaneous eye movements in the light). It can be appreciated from this figure that LL is obeyed with remarkable precision, as the standard deviation of the torsional component of monkey eye orientations is typically less than one deg (torsional range in [-2.0, +2.0] deg; Fig 3C),

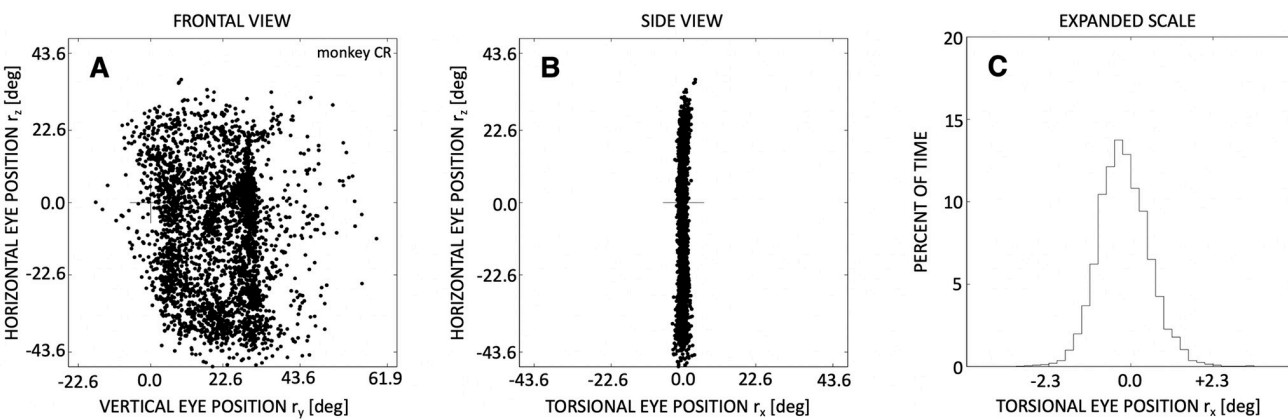

**Fig 3. Listing's law for ±3500 eye positions during spontaneous saccades in the light, of a monkey.** All eye orientations lie in a well-defined plane, which is perpendicular to the torsional direction, and defines the primary position (cross at (0,0,0)). The center of the oculomotor range is, for this animal, about 20 deg down from primary position, and the animal's head was pitched downward by 15 deg. (A) Frontal view of LP. (B) Side view of LP. (C) Expanded side view of LP. After: [18].

whereas the oculomotor range for the horizontal and vertical gaze directions runs between ±30–40 deg.

The literature has heavily debated the question whether these behavioural rules are due to a neural control strategy in the brain (e.g. [19–23]), or whether they are fully determined by the mechanical properties of the eye plant (e.g., [24–26]). For example, the pulling directions of the muscles, which eventually determine the eye's equilibrium orientation, could change in a particular way with the gaze direction, and automatically comply with Listing's law [24–26]. Certain filaments attached to the globe and surrounding the muscular tissue would function as pulleys [25], which not only prevent the muscles from excessive side-slip along the globe during eccentric viewing [27], but could also serve to mechanically implement Listing's law [24–26]. In support of this, [28] and [29] demonstrated that microstimulation of the abducens nerve, which innervates the LR muscle, generates eye movements that comply with Listing's law. These results indicate that, at least for the horizontal muscle system, the LR/MR pulleys aid in the mechanical implementation of Listing's law.

Clearly, mechanical and geometric properties of the extra-ocular muscles largely determine the direction of passive and active forces acting on the eye, and therefore on its instantaneous orientation. Note, however, that LL is only valid for eye movements with the head erect and gazing at far targets. For vestibular-evoked eye movements [30], head-unrestrained gaze shifts [31, 32], near viewing [33, 34], and microstimulation in several premotor brainstem areas [18], Listing's law is violated, and Donders' law applies. Although it has been hypothesized that the muscle pulleys could be actively controlled to shift the muscle's action dynamically, so as to accommodate programmed violations of Listing's law [35, 36], neurophysiological evidence [37] as well as theoretical arguments [38] have rendered this hypothesis unlikely. Further, it should also be noted that the saccadic system is not perfect: often, the eye spontaneously jumps out of Listing's plane by a small saccade, leading to >2 deg of cyclotorsion (e.g., Fig 3C). Interestingly, the next spontaneous saccade is then endowed with an opposite torsional component that typically brings it closer to the plane, thus preventing the eye from walking away from LP during free gazing behaviour [23]. This indicates that saccades have three, rather than two degrees of freedom, and that the control of ocular cyclotorsion may possibly involve the cerebellum [18].

In summary, regardless the underlying structural mechanics, any system that has to deal with sequences of 3D rotations has to account for noncommutativity [3]. As a result, Donders' law, and possibly also Listing's law, has to be embedded in a neural control strategy [2, 3, 18, 19, 23, 37, 39–42].

In our robotic prototype, the pulleys were kept head-fixed, so that a potentially active pulley mechanism is not considered. Instead, we here analyze the potential relationship between the emergence of Listing's law and the properties of the oculomotor plant on the basis of controlling a simple physical model.

## Main sequence nonlinearity

Apart from the restrictions on the eye's cyclo-torsion, primate saccadic eye movements also display a number of characteristic dynamic behaviors. First, saccades obey stereotyped kinematic relations, known as the main-sequence [44]: saccade duration increases with amplitude, and the peak eye velocity saturates at large saccade amplitudes. Further, because the acceleration time of saccades is approximately constant (Fig 4A), the increase in saccade duration is primarily due to its deceleration phase, and may betray an internal dynamic feedback mechanism [45]. As a result, the skewness of saccade velocity profiles increases with saccade duration [43]. These properties are summarized as follows:

$$
\begin{aligned}
\text{duration}: \quad D &= a\Delta R + b \\
\text{peak velocity}: \quad V_{pk} &= V_0(1 - \exp(-\Delta R/\alpha)) \\
\text{acceleration}: \quad T_{accel} &\approx 20 \text{ ms} \\
\text{skewness}: \quad S &= cD + d \\
\text{combined}: \quad V_{pk}D &= k\Delta R \quad \text{and} \quad V_{pk} = \frac{kc}{S - d}\Delta R
\end{aligned}
\tag{1}
$$

where $D$ is saccade duration; $\Delta R$: saccade amplitude; $V_{pk}$: peak eye velocity; $T_{accel}$: time to reach $V_{pk}$; $S$: skewness of the velocity profile, and $a, b, V_0, \alpha, c, d, k$ parameters. The two final relations hold for fast, as well as slow (drug-affected, fatigued, non-visual) saccades (Fig 4B and 4C). Together, the relations of Eq (1) indicate the presence of a nonlinear dynamic (feedback) controller in the saccadic system [46]: for a linear controller, saccade duration and shape

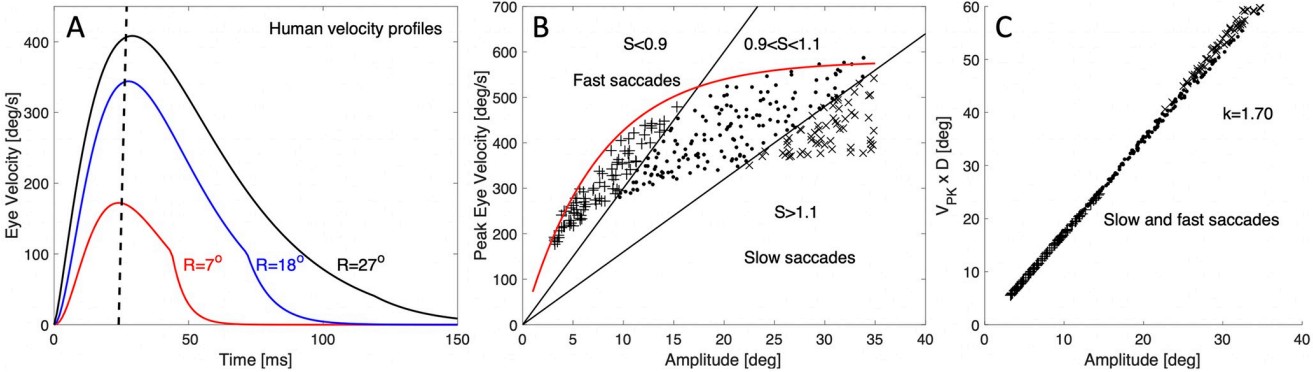

**Fig 4. (A) Example velocity profiles for saccades of different amplitudes.** Note that their peak velocity is reached at the same moment. (B) The amplitude-peak velocity relation (ordinate: 0–700 deg/s) for fast and very slow (diazepam-affected and fatigued) saccades (different symbols). The red curve delineates the fastest saccades possible for this human subject. Oblique lines separate sectors where saccades have a particular skewness. The lower the slope, the slower the saccades, and the larger its skewness. (C) Tight relation for saccades between amplitude and the product of peak velocity and duration. This relation expresses the fact that typical eye-saccades have single-peaked, 'triangular-shaped' velocity profiles. After: [43].

should be independent of amplitude, and the peak eye velocity should increase proportionally with amplitude. Because the oculomotor neurons are known to carry these properties in their bursting activity, the main-sequence behavior is not explained by a nonlinearity in the plant. Models of the saccadic system have therefore proposed nonlinear controllers at the level of the saccade-velocity activity carried by burst-neurons in the pons (for horizontal saccades [45]), and riMLF (for vertical/torsional saccades [41]).

## Straight trajectories

Oblique saccades have approximately straight trajectories, which means that the horizontal, vertical (and torsional) velocity components are scaled versions of each other. Thus, if the signal to the plant for a saccade with amplitude $\Delta R$, is encoded as a 3D angular velocity command, $\boldsymbol{\omega}_{\Delta R}(t) = [\omega_x(t), \omega_y(t), \omega_z(t)]^T$ (see below), the following holds:

$$
\begin{aligned}
\text{Hor}: \quad & \omega_z(t) = \alpha \|\boldsymbol{\omega}_{\Delta R}(t)\| \\
\text{Vert}: \quad & \omega_y(t) = \beta \|\boldsymbol{\omega}_{\Delta R}(t)\| \\
\text{Tors}: \quad & \omega_x(t) = \gamma \|\boldsymbol{\omega}_{\Delta R}(t)\|
\end{aligned}
\tag{2}
$$

with $\alpha^2 + \beta^2 + \gamma^2 = 1$.

However, because of the saturating nonlinearity in the amplitude-velocity relation (Fig 4B), the cross-coupling in the component velocity profiles is not trivial if the horizontal, vertical and torsional velocity generators would be independently driven by their own dynamic displacement errors, $\Delta z$ and $\Delta y$, and $\Delta x$, respectively:

$$
\begin{aligned}
\omega_z(t) &= f(\Delta z(t)) \\
\omega_y(t) &= g(\Delta y(t)) \\
\omega_x(t) &= h(\Delta x(t))
\end{aligned}
\tag{3}
$$

with $f(\cdot)$, $g(\cdot)$ and $h(\cdot)$ nonlinear.

For example, if a $\Delta z = 10$ deg horizontal component is part of an oblique saccade in LP, given by $[\Delta x, \Delta y, \Delta z] = [0, 20, 10]$ deg, its duration (and velocity shape) matches the larger 20 deg upward vertical component, and therefore has a longer duration (and skewness) than a purely horizontal $[0, 0, 10]$ deg saccade. This phenomenon is called component stretching, and constitutes yet another nonlinear property of the saccadic system.

However, if the component velocity generators would all be driven by a common vectorial velocity command, $\boldsymbol{\omega}_{\Delta R}(t)$, a simple trigonometric scaling of this signal ensures straight saccades and component stretching. This scheme has been introduced as the common-source model by [47].

Interestingly, neurophysiological evidence has suggested that the main-sequence nonlinearity (Fig 4B) may reside upstream from the saccade-component velocity generators, at the level of the midbrain Superior Colliculus (SC). The SC could thus serve as the common nonlinear vector-velocity generator for saccades [48, 49]. Furthermore, microstimulation at the SC motor map does not lead to violations of Listing's law [22], which suggests that the law is implemented downstream from the stimulation site, either involving brainstem-cerebellar neural circuitry [2, 18], or at the level of the oculomotor plant (see above; [24–26]).

The primary objective of the present paper is to investigate whether a feedforward optimal control strategy for saccades, applied to a simplified 3D model of the eye, as illustrated in Fig 2, could account for all observed kinematic properties of saccades (Listing's Law, main-sequence nonlinearity, and straightness of the trajectories), and how the extra-ocular muscle geometry relates to Listing's law and the orientation of Listing's plane. Although several studies have

investigated potential optimal control principles that could underlie the generation of saccades [50–52], these studies were all confined to the control of horizontal gaze shifts, and therefore did not address the 3D kinematics and dynamic problems that were central to the present study.

## Methods

### 3D eye orientation

When an extra-ocular muscle contracts to exert a force, a torque is applied on the eyeball, which causes it to rotate around its center about an axis that is perpendicular to the plane defined by the force vector and the moment arm from the eye's center to the muscle insertion point. To describe changes in eye orientation in 3D, it is convenient to define the primary position as the origin of a head-fixed right-handed reference frame, from which any new eye orientation can be reached by a single-axis rotation. In this frame of reference, a positive torsional rotation (roll) is clockwise (rotation about the frontal $+x$ axis), a positive vertical rotation (pitch) is downward (lateral $+y$ axis), and a positive horizontal rotation (yaw) is leftward (vertical $+z$ axis). There are many different ways to represent 3D rotations, but the most frequently employed ones in the oculomotor literature are unit quaternions and so-called Euler-Rodrigues rotation vectors (e.g., [6]). Here we will only summarize their relevant properties. For details, the reader is referred to [6].

Briefly, any single-axis rotation of a rigid body around the unit axis $\hat{\mathbf{n}} = [n_x, n_y, n_z]^T$ by angle $\rho$ can be parametrized by a unit quaternion (a 4-parameter complex algebraic object with length one):

$$q = \cos(\rho/2) + \sin(\rho/2)\hat{\mathbf{n}} \cdot \mathbf{I} \equiv q_0 + \mathbf{q} \cdot \mathbf{I} \tag{4}$$

with $q_0 = \cos(\rho/2)$ the (real) quaternion's scalar part, $\mathbf{I} = [i, j, k]$ are the imaginary numbers for the $(x, y, z)$ coordinates satisfying the relations $ij = k$, $jk = i$, $ki = j$, $i^2 = j^2 = k^2 = ijk = -1$, and $\mathbf{q} = \sin(\rho/2) [n_x, n_y, n_z]^T \equiv [q_x, q_y, q_z]^T$ its (imaginary) vector part. Consequently, multiplying two quaternions, $p$ and $q$, yields: $pq = p_0q_0 - \mathbf{p} \cdot \mathbf{q} + [p_0\mathbf{q} + q_0\mathbf{p} + \mathbf{p} \times \mathbf{q}] \cdot \mathbf{I}$.

It is convenient to introduce the Euler-Rodrigues formulation [16, 17], here denoted as the rotation vector, by:

$$\mathbf{r} \equiv \frac{\mathbf{q}}{q_0} = \tan(\rho/2)\hat{\mathbf{n}} \tag{5}$$

with

$$\mathbf{r}^{-1} = -\mathbf{r} \ \text{(inverse)}$$
$$\mathbf{r}_0 = \mathbf{0} \ \text{(primary position, P)} \tag{6}$$

In this latter parametrization, Donders' law and Listing's law (see above) have the following formulations:

$$\text{Donders' law}: \ r_x = f(r_y, r_z)$$
$$\text{Listing's law}: \ r_x = 0 \tag{7}$$

where the latter defines Listing's plane, expressed in the primary frame of reference (see Fig 3B).

It can be shown that the single-axis rotation vector from an initial eye position, $\mathbf{r}_A$, to any other final eye position, $\mathbf{r}_B$, is then described by the angular-velocity saccade vector (up to

$\mathcal{O}(\rho^3))$ [17]:

$$\mathbf{s}_{AB} \approx \mathbf{d} + \mathbf{r}_A \times \mathbf{d} \tag{8}$$

with $\mathbf{d} \equiv \mathbf{r}_B - \mathbf{r}_A$ (difference vector). Thus, if the initial and final eye positions are both in Listing's plane ($r_{A,x} = r_{B,x} = 0$):

$$s_{AB,x} = r_{A,y}d_z - r_{A,z}d_y = |\mathbf{d}|r_A^\perp \neq 0 \tag{9}$$

whenever angle $(\mathbf{r}_A, \mathbf{d}) \neq \{0, \pi\}$.

From Eq 9, the angular-velocity axis of the eye, $\omega \parallel \mathbf{s}_{AB}$, can be tilted out of LP by any $\eta \in [-\rho/2, +\rho/2]$, depending on the angle between the saccade difference vector, $\mathbf{d}$ and the initial eye orientation, $\mathbf{r}_A$. This geometric property is known as the half-angle rule [39].

Finally, from the quaternion relation between the angular velocity, $\omega \equiv \boldsymbol{\omega} \cdot \mathbf{I}$, the eye orientation, $q$, and the rate-of-change of eye orientation, $\dot{q}$ (the coordinate velocity; e.g. [19]):

$$\dot{q} = \frac{\omega q}{2}$$
$$\omega = 2\dot{q}q^{-1} \tag{10}$$

Expressed as rotation vectors, it follows that, up to $\mathcal{O}(\rho^3)$:

$$\dot{\mathbf{r}} \approx \frac{1}{2}(\boldsymbol{\omega} + \boldsymbol{\omega} \times \mathbf{r} + (\boldsymbol{\omega} \cdot \mathbf{r})\mathbf{r})$$
$$\boldsymbol{\omega} \approx 2(\dot{\mathbf{r}} + \mathbf{r} \times \dot{\mathbf{r}}) \tag{11}$$

The rotation vector has some nice algebraic properties, which may also be useful for neural representations. E.g., the trajectory of a single-axis rotation for a saccade in LP from A to B is well approximated (up to $\mathcal{O}(\rho^3)$) by a straight line in LP between the initial and final eye-position rotation vectors [17]:

$$\mathbf{r}(t) \approx \mathbf{r}_A + \sigma(t)\mathbf{d} \tag{12}$$

where $\sigma(t) \in [0, 1]$ (monotonic increase).

## The eye-plant model

The methodology followed in this paper can be summarized as follows. First, we created a non-linear simulator model in Matlab's Simulink, which implemented Euler's equations for rotations of a rigid body, and the noncommutative 3D rotational kinematics (described above) to our simplified mechanical model of the eye (see Fig 2). Input to the system is a time sequence of signals from three independent motors, $\mathbf{u}(t)$, that each couple in a feed-forward way to the three antagonistic pairs of eye muscles, modelled as elastic tendons. The model's output is the instantaneous rotation vector of the model eye, $\mathbf{r}(t)$, that parametrizes its current 3D orientation. To identify a linear approximation for this system for the optimal control algorithms described below, the non-linear model equations were run through Matlab's system-identification algorithms with independent pseudo-random binary signal sequences (PRBS). PRBS is a signal often used in systems identification. It is a deterministic signal composed of rectangular pulses of fixed amplitude but variable duration that has similar properties to white noise (flat spectrum), thus excites equally all the frequencies of interest in the system. The amplitude range of these signals was chosen such as to obtain good approximations of the model's output on the range of the desired saccade amplitudes. The identified model was then used to compute the optimal input sequences to control saccades, by applying different sets of

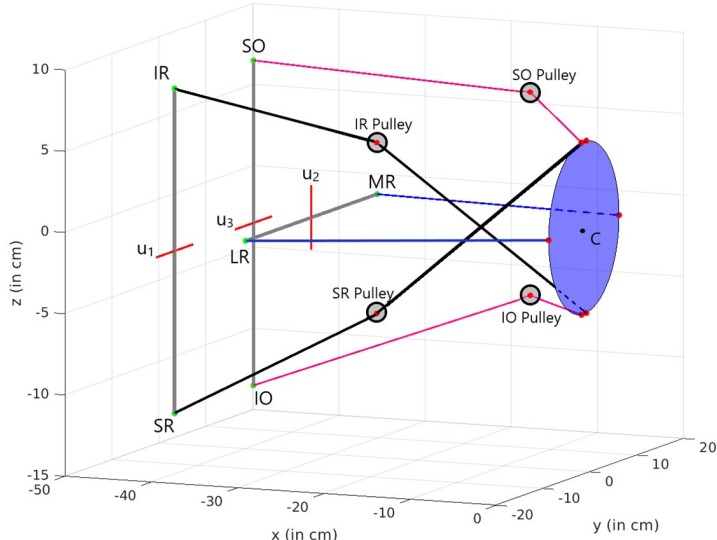

**Fig 5. Schematic of the mechanical model of Fig 2.** The common driving signals, $u_{1,2,3}$, are symbolized by the grey bars, that rotate around the red axes, thus providing an antagonistic contraction/relaxation force on the associated muscles (cables with same colours) from their end-insertion points at $\mathbf{P}_i$ (green dots). The six muscle insertion points, $\mathbf{Q}_i$, on the eye (red dots) form the (elliptical) blue surface; $C$ is the center of the globe at $(0,0,0)$. The default arrangement is nearly symmetrical with respect to the horizontal plane of the eye at $z = 0$.

combined performance costs. Finally, the optimized inputs were provided to the nonlinear simulator to generate 1500 randomly distributed simulated eye movements, of which we analyzed the kinematic and dynamic properties.

**The simulator.** To model the eye we did not attempt to fully replicate the anatomical and physiological features of the individual muscles. Instead, we only included their most essential features: their insertion points on the globe and approximate pulling directions, their elastic properties (note that the pull is caused by a lengthening of the string, rather than by a physiological contractile element; the mechanical effect on the eye's orientation, however, is identical), dynamic friction (a viscous force, caused by the drag of the optic nerve and orbital tissues through the fatty layers), and the way in which the muscles will change their direction of action when the eye orientation changes.

The input for the simulator (the angular positions of the drivers) was provided as a 3D command $\mathbf{u}(t) = [u_1(t), u_2(t), u_3(t)]^T$, which acted as three independent signals for the three antagonistic muscle pairs (see Fig 5).

The model's output corresponded to the resulting rotation vector, $\mathbf{r}(t) = [r_x(t), r_y(t), r_z(t)]^T$, which represents the 3D orientations of the eye that produced the resulting rotational movement trajectory.

The rotational dynamics for this rigid-body mechanical system, expressed in the inertial frame, are governed by Euler's equation:

$$
\begin{aligned}
\boldsymbol{\tau}_T &= \sum_i \boldsymbol{\tau}_{i,el} + \boldsymbol{\tau}_{fric} = \\
&= I(q)\boldsymbol{\alpha} + \boldsymbol{\omega} \times (I(q)\boldsymbol{\omega})
\end{aligned}
\tag{13}
$$

where $\boldsymbol{\tau}_T$ is the total torque on the eye, $\boldsymbol{\tau}_{i,el}$ are the elastic torques applied by the six muscles to the eye at their insertion points, $\mathbf{Q}_i$, and $\boldsymbol{\tau}_{fric}$ is the total frictional torque felt by the eye (viscosity, $\boldsymbol{\tau}_{dyn}$ and static friction, $\boldsymbol{\tau}_{stat}$, see below; we ignored the force of gravity); $I(q)$ is the eye's

moment of inertia tensor (which may change as the eye rotates about an arbitrary axis around its center, $C$, and the mass distribution with respect to the primary reference frame changes too), $\boldsymbol{\alpha}$ is the 3D instantaneous angular acceleration vector (in rad/s$^2$), and $\boldsymbol{\omega}$ is the eye's angular velocity (in rad/s). The cross-product term vanishes when $I(q) = aI_3$ (where $I_3$ is the 3x3 unity matrix). For our simulator, we verified that its contribution was negligible when compared to the angular acceleration term, as the relative rms power of $|\boldsymbol{\omega}(t) \times (I\,\boldsymbol{\omega}(t))|$ vs. $|I\boldsymbol{\alpha}(t)|$ remained $<2 \cdot 10^{-4}$ for saccade amplitudes up to 60 deg (S1(D) Fig). We therefore neglected the cross-product term in the simulator.

The torques were calculated from the simplified geometry, illustrated in Fig 5. Table 1 provides the coordinates of the insertion points in the $(x, y, z)$ coordinate frame of the model's eye ball (re. its center), the coordinates of the via points, $\mathbf{X}_i$, for the pulleys of the SR, IR, SO and IO muscles (the LR and MR were directly connected to their muscle endings at $\mathbf{P}_i$), and the resulting normalized torques exerted by the muscles when the eye is at its equilibrium orientation, $\mathbf{r} = (0,0,0)$. The insertion points at which the driving signals $\mathbf{u}$ exercise their action are represented by $\mathbf{P}_i$. The changes of the insertion coordinates, from $(P_{i,0,x}, P_{i,0,y}, P_{i,0,z})$ (Table 1) to $(P_{i,x}, P_{i,y}, P_{i,z})$, of the vertical recti and obliques ($i = $ [SR,IR], or [SO, IO]), which are controlled by $u_1$ and $u_3$, respectively, in radians) were determined by the following relations:

$$
\begin{aligned}
P_{i,x} &= P_{i,0,x} \pm \frac{L_M}{2}\sin{(u_{1,3})} \\[4pt]
P_{i,y} &= P_{i,0,y} \\[4pt]
P_{i,z} &= P_{i,0,z} \pm \frac{L_M}{2}(1 - \cos{(u_{1,3})})
\end{aligned}
\tag{14}
$$

with $L_M = 20$ cm. For the lateral recti ($i = $ [LR,MR], with $L_R = 28$ cm, controlled by $u_2$) this yielded:

$$
\begin{aligned}
P_{i,x} &= P_{i,0,x} \pm \frac{L_R}{2}\sin{(u_2)} \\[4pt]
P_{i,y} &= P_{i,0,y} \pm \frac{L_R}{2}(1 - \cos{(u_2)}) \\[4pt]
P_{i,z} &= P_{i,0,z}
\end{aligned}
\tag{15}
$$

The locations of the muscle insertion points on the eye, specified at $\mathbf{Q}_{0,i}$ in the primary position (Table 1), will change as a result of the eye's rotation (given by the rotation matrix, $R$)

**Table 1. Coordinates of the insertion points (in cm) on the mechanical model of the right eye at rest ($\mathbf{Q}_{0,i}$), and the craniocentric via points ('pulleys') ($\mathbf{X}_i$) with respect to the (head-fixed) center of the eye.** Here, LR and MR have no pulleys; they directly connect to driver 2. The cranial insertion points, $\mathbf{P}_{i,0}$ at rest are given in columns 8–10. Columns 11–13: the unit torque-vector components, as computed from Eq (18) with the eye in the resting orientation. Note that the SO, IO, SR and IR tendons all pull with a prominent cyclo-torsional ($x$) and vertical ($y$) component.

| Muscle | $Q_{0,x}$ | $Q_{0,y}$ | $Q_{0,z}$ | $X_x$ | $X_y$ | $X_z$ | $P_{0,x}$ | $P_{0,y}$ | $P_{0,z}$ | $\tau_x$ | $\tau_y$ | $\tau_z$ |
|--------|-----------|-----------|-----------|-------|-------|-------|-----------|-----------|-----------|----------|----------|----------|
| MR | -0.7 | 7.5 | 0 | - | - | - | -32.3 | 14.0 | 0.5 | -0.01 | 0.00 | 1.00 |
| LR | -0.7 | -7.5 | 0 | - | - | - | -32.3 | -14.0 | 0.5 | 0.01 | 0.00 | -1.00 |
| SR | -0.7 | 0.53 | 5.3 | -20.6 | -7.5 | -5.3 | -43.6 | -8.4 | -7.9 | 0.35 | -0.93 | -0.05 |
| IR | -0.7 | 0.53 | -5.3 | -20.6 | -7.5 | 5.3 | -43.6 | -8.4 | 12.1 | -0.31 | 0.94 | 0.13 |
| SO | -0.7 | -0.53 | 5.3 | -15.0 | 15.0 | 6.3 | -43.6 | 8.4 | 12.1 | -0.71 | -0.70 | -0.02 |
| IO | -0.7 | -0.53 | -5.3 | -15.0 | 15.0 | -6.3 | -43.6 | 8.4 | -7.9 | 0.73 | 0.66 | -0.16 |

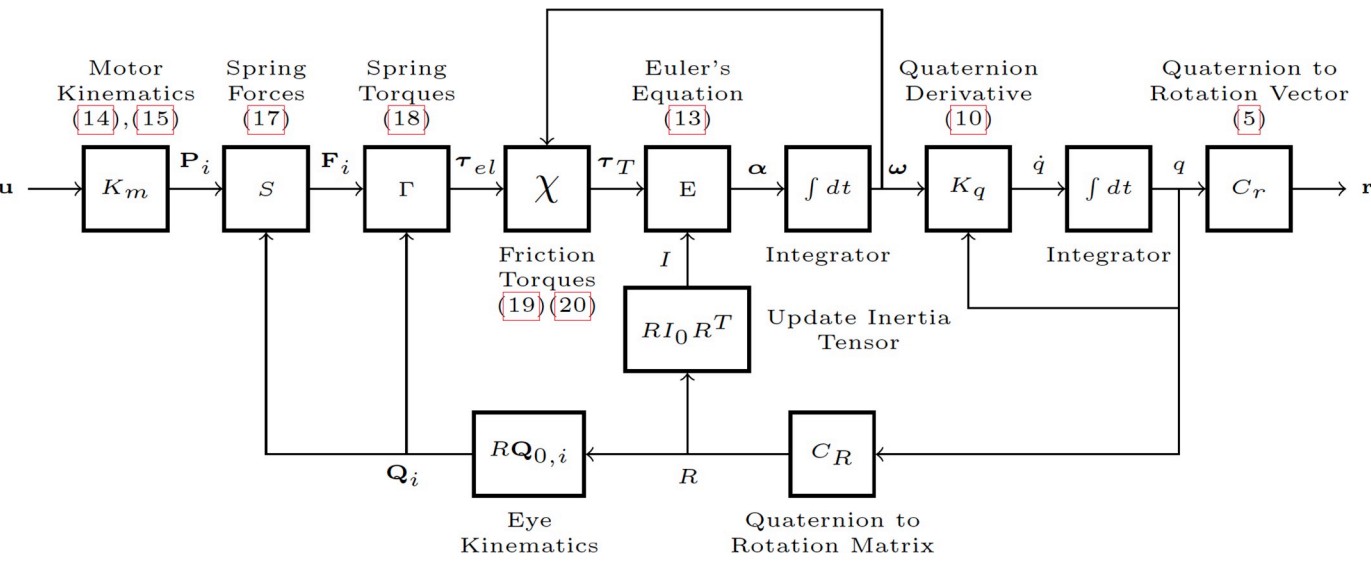

**Fig 6. Block diagram of 3D eye saccade simulator.** Close to each block is a summary description and the equation numbers that implement the associated functions (when applicable).

according to

$$\mathbf{Q}_i = R\mathbf{Q}_{i,0} \tag{16}$$

Because of the changes in the eye-insertion points, the muscles will change their lengths and exert elastic forces $\mathbf{F_i}$ through Hooke's law:

$$\mathbf{F}_i = -\kappa_i(l_i - l_{0,i})\hat{\mathbf{t}}_i \tag{17}$$

with $l_{0,i}$ the rest length of muscle $i$, and $l_i \geq l_{0,i}$ the current length of the muscle. $\kappa_i$ is the elasticity of the muscle, and was given the same value of 6.0 N/m for all tendons. $\hat{\mathbf{t}}_i$ is the unit vector of the tendon's pulling direction from $\mathbf{Q}_i$ to $\mathbf{X}_i$ for SO, IO, SR and IR, and to $\mathbf{P}_i$ for ML and LR. The elastic torques (Fig 6 and Table 1) are then determined by

$$\boldsymbol{\tau}_{i,el} = \mathbf{Q}_i \times \mathbf{F}_i \tag{18}$$

Besides the elastic forces, the eye also feels frictional forces, because of its rotation through the fatty embedding within the orbit. In the model, we included both a small static friction (by Newton's 3rd law: equal and opposite to the total exerted torque) and an angular-velocity dependent dynamic friction. The dynamic frictional force was approximated by:

$$\boldsymbol{\tau}_{dyn} = -\chi_{dyn}\boldsymbol{\omega} \tag{19}$$

where $\chi_{dyn}$ = 0.02 Nms/rad is the dynamic friction coefficient. This parameter, which relates to the viscosity of the plant, was set to make the eye behave as an over-damped system.

For the static friction we took:

$$\boldsymbol{\tau}_{stat} = \begin{cases} 0 & \Leftarrow |\boldsymbol{\omega}| > \epsilon \vee |\boldsymbol{\tau}_{comp}| \geq \chi_{stat} \\ -\boldsymbol{\tau}_{comp} & \Leftarrow |\boldsymbol{\omega}| < \epsilon \wedge |\boldsymbol{\tau}_{comp}| < \chi_{stat} \end{cases} \tag{20}$$

with $\boldsymbol{\tau}_{comp} = \boldsymbol{\tau}_{el} + \boldsymbol{\tau}_{dyn}$, $\epsilon$ a small dynamic cut-off velocity, $|\boldsymbol{\omega}|$ is the amplitude of the angular velocity vector, and $\chi_{stat}$ = 0.006 Nm is the static friction coefficient.

Once all torques are known, the angular acceleration, $\boldsymbol{\alpha}$, can be computed through Eq (13), given that the inertial tensor can be assumed constant for each small time step of the simulation ($\Delta t = 10$ ms). The angular velocity vector of the eye, $\boldsymbol{\omega}$ (and hence, its instantaneous axis of rotation, $\hat{\mathbf{n}}$, Eq (5)) is obtained from the angular acceleration by time integration. The resulting new orientation of the eye, $q$ (or, alternatively, its rotation vector, $\mathbf{r}$; Eq (4)), is subsequently computed by integrating the coordinate-velocity $\dot{q}$, which was obtained from $q(t - \Delta t)$, and $\boldsymbol{\omega}$ from Eq (10). When the new orientation, $q(t)$, is calculated, it is used to update the coordinates of the insertion points on the eye $\mathbf{Q_i}$ and to update the tensor of the moment of inertia by $I(R) = RI_0 R^T$ (with $I_0$ the moment of inertia tensor in the eye's primary reference frame, and $R$ the rotation matrix associated with the quaternion $q$; see Fig 6). For a perfect symmetric sphere $I_0$ would be given by $I_{sphere} = \frac{2}{5}ma^2$, with $a$ its radius, and $m$ the mass, and invariant to eye orientation. In our robotic prototype, it is given by:

$$I_0 = \begin{bmatrix} 4.759 & -0.01 & 0.111 \\ -0.01 & 4.316 & 0 \\ 0.111 & 0 & 3.956 \end{bmatrix} \cdot 10^{-4} \quad \text{kg} \cdot \text{m}^2 \tag{21}$$

A simplified diagram of the simulator is shown in Fig 6. The simulator was implemented in Matlab's Simulink toolbox (version 2018b). It is important to note that the equations governing the simulator are nonlinear. This is already obvious from the nonlinear updates of the cranial insertion points (Eqs (14) and (15)). However, also the pulling directions of the elastics change as the eye rotates, causing the gains between inputs and outputs to depend on the operating point. Moreover, the 3D kinematic relations include vector products which are non-commutative, and contain trigonometric dependencies on the angles. This all means that the model can only be well approximated by a linear system around a given operating point, and outside that region is expected to produce large errors.

To check whether the model was mechanically capable to cover the full range of 3D eye movements, we simulated a large feedforward set of 21 rotational steps for each of the three motors over a range of [-35, +35] deg (i.e., a total of $21^3 = 9216$ combinations), to evaluate the 3D oculomotor range of the model. Fig 7 shows the result of this simulation. Clearly, the horizontal, vertical, and torsional ranges of the model eye were quite comparable, and close to [-50, +50] deg in all directions.

**Linear system identification.** The system identification procedure [53, 54] consisted of the following steps. As input for the identification we used a 180 s long Pseudo-Random Binary Sequence (PRBS) signal, uncorrelated between the three motor inputs. The first 120 s were used to train the system-identification algorithm (typically yielding >98% accuracy for all components), while the remaining 60 s of the data were used for testing the model predictions and validating the quality of the fit. We constructed a state-space model from our system, which represents the relation between its inputs and outputs through a set of $n^{th}$ order differential equations, comprised within a single first order $n \times n$ matrix difference equation. We consider that the dominant dynamics is given by the eye mechanical inertia (the motor electric and mechanical time constants are comparatively much smaller), so the state vector has intrinsically 6 dimensions (3 orientations and 3 angular velocities). In this way, the discrete dynamical system is represented by the following two matrix equations,

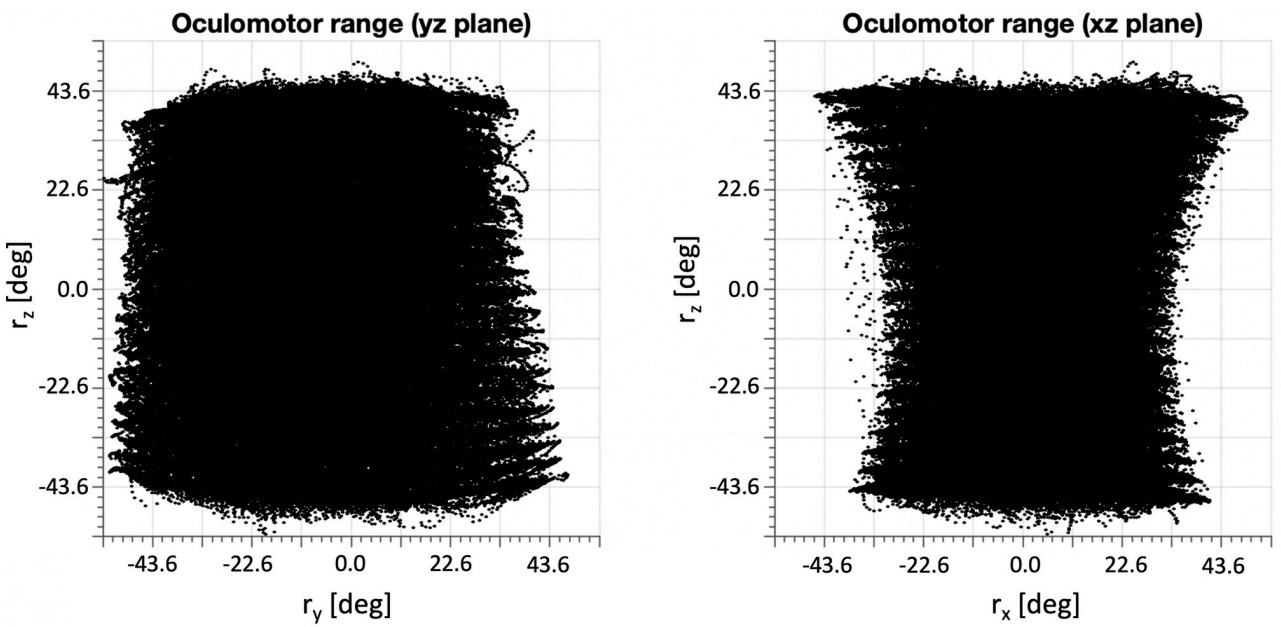

**Fig 7. 3D oculomotor range of the model in two planar views (yz-plane, left and xz-plane, right), when the inputs ran from [-35, +35] deg for each of the three muscle pairs.** Note that the torsional range of the model is substantial, and quite comparable to the horizontal and vertical ranges.

with $n = 6$:

$$\mathbf{x}_{t+1} = A\mathbf{x}_t + B\mathbf{u}_t$$
$$\mathbf{r}_t = C\mathbf{x}_t + E\mathbf{u}_t \tag{22}$$

where $t$ is the discrete time variable. To find the optimal parameters, the matrices [$A$, $B$, $C$, $E$], of the model, we applied the subspace method ([55]; implemented in Matlab's "n4sid.m" function), which is known to provide numerically stable state-space models for multivariable dynamical systems [54, 56].

To quantify the quality of the identification, we calculated the normalized root-mean squared error (NRMSE) fitness value, which indicates how well the predicted model response matches the non-trained target data for the nonlinear simulation:

$$NRMSE(\%) = 100\left(1 - \frac{\|\mathbf{r} - \hat{\mathbf{r}}\|}{\|\mathbf{r} - \bar{\mathbf{r}}\|}\right), \tag{23}$$

where $\mathbf{r}$ is the true validation data output of the nonlinear system, $\hat{\mathbf{r}}$ is the output of the identified linear system and $\bar{\mathbf{r}}$ is the mean of $\mathbf{r}$.

Even though the predictions on the validation set for the linearized fit of Eq (22) were not perfect, the result was generally judged satisfactory. The torsional and horizontal components exhibited NRMSE values above 75% for the torsional component, and 90% or higher for the horizontal component, which is excellent. The vertical component, however, lagged somewhat behind with a best correlation of 40%. It should be noted, however, that the linearization will always produce errors, since the simulator is highly non-linear, especially for the strongly cross-coupled torsional/vertical muscle pairs. Indeed, correlations for $r_x$ and $r_y$ increased substantially, when the via points for SO and IO were positioned less laterally (not shown).

**Optimal control.** We next used the identified system, Eq (22), to find the optimal control commands $\mathbf{u}_i$ that minimize a total cost functional. Several cost functions for the gaze control

system have been proposed in past studies to account for the nonlinear main-sequence properties of Eq (1). An optimal control strategy that minimizes saccade duration and maximizes saccade accuracy has proved successful (speed-accuracy trade off; e.g. [50]). However, models on optimal gaze control have so far been confined to one dimension only (horizontal saccades), and therefore miss other important aspects of gaze control, like component cross-coupling, and Listing's law. Here, we evaluated the contribution of different cost terms on overall saccade performance.

In general, the optimal open-loop control problem is represented as the solution to the following constrained optimization problem:

$$\text{minimize }_{\mathbf{U},D} \quad J_{TOT} = \sum_\alpha \lambda_\alpha J_\alpha(\mathbf{U}, D, \mathbf{R})$$

$$\text{subject to}: \quad \mathbf{x}_{t+1} = A\mathbf{x}_t + B\mathbf{u}_t \tag{24}$$

$$\mathbf{r}_t = C\mathbf{x}_t + E\mathbf{u}_t \qquad t = 0, ..., D$$

where $J_{TOT}$ is the total cost, and $[\lambda_\alpha, J_\alpha]$ represent the relative weight and costs of each of the terms that contribute to $J_{TOT}$. $D$ is the saccade duration in discrete time steps, and $\mathbf{U} = [\mathbf{u}_0, \mathbf{u}_1, \ldots \mathbf{u}_D]$, and $\mathbf{R} = [\mathbf{r}_0, \mathbf{r}_1, \ldots \mathbf{r}_D]$ are the time series of inputs and outputs, respectively, from the beginning of the saccade (at $t = 0$) until its offset at $t = D$. The constraint is applied to the linearized identified model of Eq (22), and parametrized by matrices $[A, B, C, E]$.

Our aim is to identify the simplest cost function (with the smallest set of cost terms) that best accounts for the following properties:

- Control of 3D saccadic movements, such that all axes describing eye orientations lie in a plane (Listing's plane), according to Eq 7

- Nonlinear dynamic properties of saccades (their main-sequence relations, including skewness, straight trajectories, and the associated component stretching of velocity profiles).

**Costs.** The first three cost terms to be considered have also been proposed in earlier studies. They acount for (i) minimization of the gaze-direction error (i.e., maximize saccade accuracy), (ii) minimize the total energy spent by the input signals, and (iii) minimize saccade duration, $D$.

**Accuracy.** The accuracy cost minimizes the mean-squared difference between the target direction, $(\hat{r}_y, \hat{r}_z)$, and the actual 2D gaze direction at the end of the saccade, $(r_{y,D}, r_{z,D})$, without putting any additional constraints on the amount of cyclo-torsion, $r_x$. Moreover, as the saccade reaches its goal, the velocity and acceleration should be zero. Thus, the classical two-dimensional accuracy cost is written as:

$$J_A(\mathbf{r}_D) = (r_{y,D} - \hat{r}_y)^2 + (r_{z,D} - \hat{r}_z)^2 \quad \text{and} \quad [\dot{\mathbf{r}}_D, \ddot{\mathbf{r}}_D] = 0 \tag{25}$$

Clearly, this cost term is essential for any optimization procedure, as without it the system's performance will not be bound to any particular input-output relationship. For some solvers, it may be better to include zero velocity and acceleration constraints in the cost function to penalize the norms of the velocity and acceleration vectors.

**Energy.** The energy expenditure by the input signals was taken to be proportional to the total squared input velocity vector across time. The latter is proportional to the instantaneous changes in the input vector, $d\mathbf{u}_t = \mathbf{u}_{t+1} - \mathbf{u}_t$. In this way, the cumulative energy cost was

expressed by the length of the time-series difference vector:

$$J_E(\mathbf{U}) = \|\Delta\mathbf{U}\|^2, \tag{26}$$

with $\Delta$ a matrix consisting of blocks of 3x3 identities, such that

$$\Delta\mathbf{U} = \begin{bmatrix} I_3 & 0 & \ldots & 0 \\ -I_3 & I_3 & \ldots & 0 \\ \vdots & \vdots & \ddots & \vdots \\ 0 & \ldots & -I_3 & I_3 \end{bmatrix} \mathbf{U} = \begin{bmatrix} \mathbf{u}_0 \\ \mathbf{u}_1 - \mathbf{u}_0 \\ \vdots \\ \mathbf{u}_D - \mathbf{u}_{D-1} \end{bmatrix}. \tag{27}$$

**Duration.** The duration term penalizes a long saccade duration, $D$. To implement a duration cost, we took the hyperbolic discount of reward at $D$, proposed by [57]:

$$J_D(D) = \left(1 - \frac{1}{1 + \beta D}\right), \tag{28}$$

with $\beta$ is the rate at which the value of reward decays.

**Listing's law.** Because the first three costs do not put any constraints on the 3D behaviour of the eye movement, we also considered costs that might enforce a behaviour according to Listing's Law (Eq (7)). We considered two costs that explicitly penalize deviations from Listing's plane: one that only incorporates the final 3D orientation of the eye, regardless the trajectory followed, and one that constrains the entire eye-movement trajectory to Listing's plane.

Thus, the cost that penalizes deviations from Listing's plane at saccade offset is determined by:

$$J_{LP,T_{off}}(\mathbf{r}_D) = \|r_{x,D}\|^2 \tag{29}$$

while constraining the entire eye movement trajectory to Listing's plane was associated with the following cost:

$$J_{LP,T_{all}}(\mathbf{R}) = \|\mathbf{R}_x\|^2 \tag{30}$$

with $\mathbf{R}_x = [r_{x,0}, r_{x,1}, \cdots, r_{x,D}]$.

**Force.** Note that the functionals of Eqs (29) and (30) assume an a-priori representation of Listing's plane in the system. As such, the primary position is explicitly assumed to correspond to the straight-ahead eye orientation. However, if Listing's plane were to be an emerging property of the optimization, such an assumption should preferably not be imposed a priori. In the default mechanical model of Fig 5, and Table 1, the origins of the extraocular muscles at the back of the orbit were arranged approximately symmetrically around the horizontal plane of the eye ($z = 0$). However, if these origins were to be jointly shifted upward or downward, the orientation of the primary position (the normal vector to Listing's Plane) might change accordingly. This possibility would suggest that the orientation of Listing's Plane could be related to an (a)symmetry in the pulling directions around the horizontal plane of the eye with respect to straight ahead. We thus included a cost that would minimize the total tension exerted by the extraocular muscles on the globe at the final eye orientation (i.e., fixation at $t = D$), and hence, would minimize the total effort required from the neural inputs when the eyes are not moving. As an empirical approximation, we related the total force and eye orientation

**Table 2. Six different cost functionals, all optimized to control saccades by tuning the weights, $\lambda_\alpha$ for the contributing terms.** Each total cost includes the accuracy cost, but differs in the contribution from other kinematic terms. AD: accuracy and duration; AE: accuracy and energy; AED: accuracy, energy and duration; AEDL$_1$: AED with Listing's law at saccade offset only; AEDL$_2$: AED with Listing's law for the entire trajectory. The weights are the (two to four) free parameters for the optimal control.

| Costs | Total movement cost |
|---|---|
| AD | $J_{AD} = \lambda_A J_A(\mathbf{r}_D) + \lambda_D J_D(D)$ |
| AE | $J_{AE} = \lambda_A J_A(\mathbf{r}_D) + \lambda_E J_E(\mathbf{U})$ |
| AED | $J_{AED} = J_{AD} + \lambda_E J_E(\mathbf{U})$ |
| AEDL$_1$ | $J_{AEDL_1} = J_{AED} + \lambda_{LP1} J_{LP,T_{off}}(\mathbf{r}_D)$ |
| AEDL$_2$ | $J_{AEDL_2} = J_{AED} + \lambda_{LP2} J_{LP,T_{all}}(\mathbf{R})$ |
| Force | $J_{Force} = J_{AED} + \lambda_F J_F(\mathbf{r}_D)$ |

by the following quadratic form:

$$J_F \approx \mathbf{r}_D^T H_F \mathbf{r}_D \tag{31}$$

This quadratic relation resulted from simulations with the model across the system's oculomotor range for different configurations of the origins and eye orientations. To estimate the $3 \times 3$ matrix $H_F$, we gathered data across a large range of 3D eye orientations, like in Fig 7.

The weight of the force cost was taken much smaller than the weight for the accuracy cost, i.e. $\lambda_F \ll \lambda_A$, because the objective was that the force minimization should only differentiate between saccades ending in the same vertical and horizontal desired orientations, and not deteriorate the accuracy of the final gaze-direction components $r_{y,D}$ and $r_{z,D}$.

Although, in principle, 32 different cost functions can be constructed from any combination of the accuracy cost with the other five, we considered the six cost functionals in Table 2 as the most relevant candidates for the optimal control.

The optimal saccade commands $\mathbf{U}$, and saccade durations, $D$ (at 10 ms resolution), were found by solving the quadratic problems with linear constraints, outlined in Eq (24) and Table 2, through applying Matlab's "quadprog.m" optimization algorithm. Fig 8 (left) shows an example of the optimization for the $J_{AED}$ cost as function of the movement duration for a 10 deg horizontal saccade. In the simulations, we ran 1500 saccades in random directions. The saccade vectors were drawn from a Gaussian distribution for the vertical ($\hat{r}_y$) and horizontal ($\hat{r}_z$) components with a standard deviation of 15 deg. The first saccade started at the (assumed) primary position, $\mathbf{r}_0 = 0$, and any subsequent saccade started from the new end position in a random direction and amplitude (Fig 8, right). On this set of randomly directed saccades, we implemented the different optimization strategies outlined in Table 2.

## Results

We characterized the 3D behaviour of the model by analyzing the saccade trajectories, kinematics, and velocity profiles. To quantify the distribution of rotation vectors, we considered all data points, $\mathbf{R}$, and analyzed their deviations from the best-fit plane. We analyzed the behavior for two different simulator configurations: one, in which the insertion points, $\mathbf{P}_i$ were symmetrical with respect to the eye's equator ($z = 0$, like in Fig 5), and one in which we systematically varied all insertions together along the $z$-direction (the asymmetrical case).

For the symmetrical simulator, the primary position is expected to coincide with the straight-ahead orientation, so that Listing's plane is simply defined by $r_x = 0$ for all eye orientations. To compare the different optimal control strategies, we analyzed the dispersion

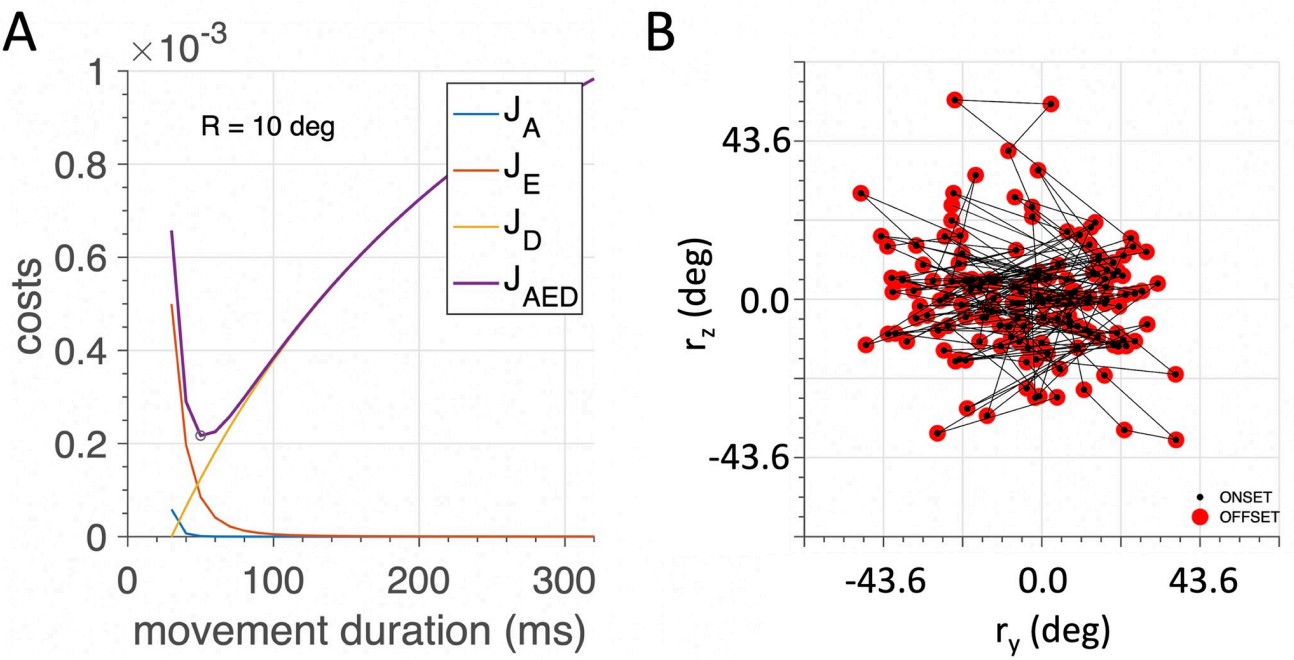

**Fig 8.** (A) Example of minimizing the total movement cost, $J_{AED}$ (purple curve), as function of the simulated saccade duration for a 10 deg rightward saccade. In this case, the optimum is reached at a duration of 50 ms (circular symbol). (B) Illustration of the first 150 goals for the saccade planner in degrees. The simulator started at the straight-ahead fixation point at (0,0), with the eye in the (assumed) primary position, at $\mathbf{r}_0 = 0$, in a random direction, and any subsequent saccade started (black dot) from the end point of its predecessor (red symbol).

(standard deviation) of the torsional component of the eye orientations with respect to the model's Listing's Plane: $\sigma_{r_x}$.

To quantify the straightness of the resulting saccades (that is, to what extent they are generated as a single-axis rotation, and follow the rules of component stretching, described

**Table 3. Results for the different optimization strategies specified in Table 2. n**: normal to the best-fit plane through the rotation-vector data, and absolute angle between **n** and the straight-ahead direction, **P** = [1, 0, 0]; Width LP: standard deviation of the data around the best-fit plane. MS: nonlinear main-sequence data fit on all saccade data: $V_0$: asymptotic velocity (deg/s); $\alpha$: angular constant (in deg); $r_{CS}$: component stretching correlation of $v_{pk,\Delta H=8^\circ}(\Phi)$ vs. $\cos(\Phi)$ (cf. Fig 11B).

| Optimization | normal **n** | re. **P** (°) | Width LP (°) | MS: [$V_0$, $\alpha$] | $r_{CS}$ |
|---|---|---|---|---|---|
| AE | [0.969, -0.245,-0.032] | 14.3 | 1.64 | [647, 61.2] | -0.04 |
| AD | [0.990, -0.137, 0.036] | 8.1 | 3.29 | [9607, 493.2] | -0.98 |
| AED | [0.955, -0.266,-0.129] | 16.5 | 5.88 | [689, 32.2] | 0.96 |
| AEDL$_1$ | [0.997, -0.073, 0.012] | 4.2 | 1.71 | [589, 20.5] | 0.82 |
| AEDL$_2$ | [1.000, -0.018, 0.004] | 1.1 | 1.59 | [443, 22.4] | 0.87 |
| Force | | | | | |
| $\Delta z = -8$ | [0.995, -0.088, 0.055] | 5.1 | 1.77 | [571, 19.7] | 0.95 |
| $\Delta z = -6$ | [0.996, -0.081, 0.032]] | 4.7 | 1.69 | [559, 18.5] | 0.96 |
| $\Delta z = -4$ | [0.997, -0.080, 0.007] | 4.6 | 1.64 | [538, 17.0] | 0.92 |
| $\Delta z = -2$ | [0.996, -0.083, -0.025] | 5.0 | 1.60 | [518, 15.4] | 0.94 |
| $\Delta z = 0$ | [0.994, -0.087, -0.062] | 6.1 | 1.55 | [500, 13.9] | 0.96 |
| $\Delta z = 2$ | [0.993, -0.040, -0.107] | 6.6 | 1.64 | [476, 12.2] | 0.97 |
| $\Delta z = 4$ | [0.986, -0.055, -0.157] | 9.6 | 1.77 | [472, 11.4] | 0.90 |
| $\Delta z = 6$ | [0.976, -0.070. -0.205] | 12.5 | 1.89 | [469, 11.1] | 0.90 |
| $\Delta z = 8$ | [0.964, -0.085, -0.251] | 15.4 | 2.03 | [464, 10.9] | 0.92 |

in the Introduction) we calculated the correlations between the vertical and horizontal velocity profiles, represented by $\bar{\rho}_{v_y, v_z}$, with $\mathbf{v} = \dot{\mathbf{r}}$. For a straight saccade, this value should be close to one.

Finally, we analyzed the main-sequence properties of the saccades: the relations between duration, peak velocity and amplitude of saccades (Fig 4), and the stretching of the vector components for oblique saccades. Because the temporal resolution of the simulations was set at 10 ms, we did not calculate a measure for the skewness of velocity profiles, as it would be too coarse. We analyzed the dynamic properties for the entire set of 1500 saccades, as well as for three selected sets of 12 saccades made in three different directions (horizontal, oblique (at a 45 deg angle) and vertical), all starting from the primary position.

In what follows, we present the results for the optimal control of $J_{AED}$ (no constraint on ocular torsion), of $J_{AEDL_1}$ (constraining the final eye orientation to Listing's Plane), and of $J_{Force}$ for nine asymmetric configurations (without putting an explicit constraint on ocular torsion). The results for the other optimization strategies (AE, AD, and AEDL$_2$) are summarized in Table 3, and in S2–S5 Figs.

## Minimizing AED

Fig 9 shows the main simulation results for the 1500 saccades resulting from the AED cost minimization.

**Listing's plane analysis.**   As can be seen in Fig 9A and 9B, the model eye looked all across the visual field (about 50° range both in horizontal and vertical gaze directions in the $yz$ plane). The best-fit plane, $r_x = ar_y + br_z$, through the 3D eye-movement trajectories had a normal vector of $\hat{\mathbf{n}} = [0.955, -0.266, -0.129]$, with the data scattering around the plane with a standard deviation of $\sigma_x = 0.043$ half-radians (4.88 deg). The best-fit plane is therefore rotated with respect to the frontal ($yz$) plane, along the $y$- and $z$-axes. This can be appreciated by the slight forward tilt of the data cloud in the ($xz$)-view (panel B), in which the torsional scatter is nearly twice as large (10.95 deg; cf. Fig 3C). Clearly, the scatter around the best-fit plane is much smaller than the torsional oculomotor range of the model, illustrated in Fig 7.

The axis and angle that will rotate the normal to the plane, $\hat{\mathbf{n}}$ (which is expressed in the eye-centered reference frame), onto the true primary direction, $\mathbf{P} = (1, 0, 0)$ of Listing's reference frame, is given by $\hat{\mathbf{n}}_\mathbf{p} = \hat{\mathbf{n}} \times \mathbf{P} = [0, -0.437, 0.900]$, and $\rho_P = \arccos(\hat{\mathbf{n}} \cdot \mathbf{P}) = 17.2$ deg. Applying the associated rotation matrix, $R_P$, to the rotation-vector data of Fig 9 then yields the eye orientations expressed in Listing's frame of reference:

$$\mathbf{r}_L = R_P \mathbf{r} \tag{32}$$

The result of this coordinate transformation is shown in Fig 10 for the ($x, z$) view.

**Trajectory analysis.**   The simulations also showed that saccade trajectories were approximately straight, as the correlations between the horizontal and vertical velocity profiles had their mode very close to 1.0 (histogram, Fig 9D). This property emerges from the simultaneous optimization of the duration and energy costs. A straight trajectory thus ensures a movement with the shortest possible duration, at the lowest energy cost.

**Main sequence analysis.**   The main-sequence plot of all saccade data is presented in Fig 9C. Note that there is a considerable variability in the peak vectorial eye velocities, which is due to the fact that saccades started all across the oculomotor range, and were pooled for all movement directions. Because of the mechanical properties of the eye muscles, it may be expected that the peak eye velocity will be position- and direction-dependent. To illustrate the

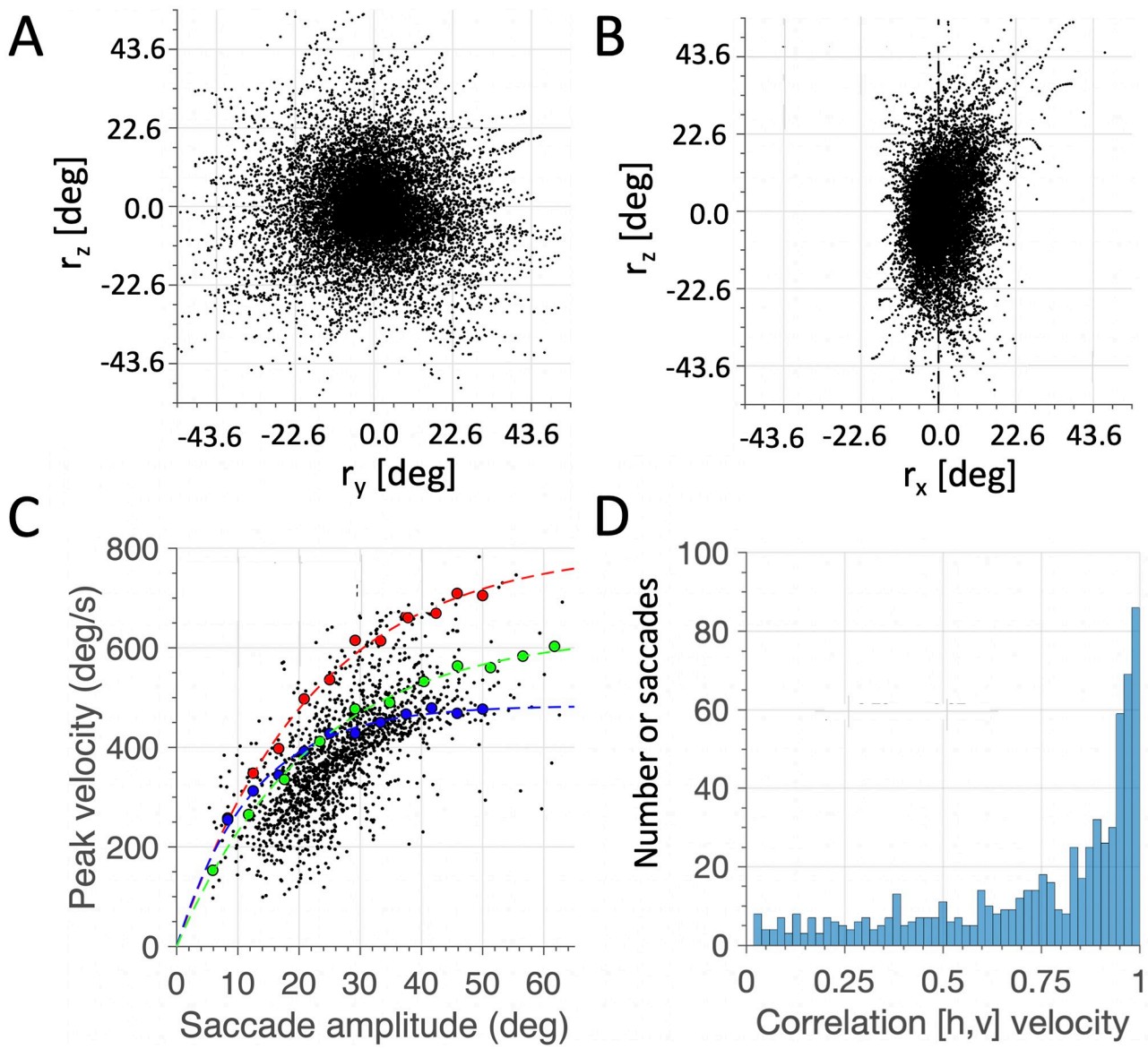

**Fig 9. 3D oculomotor behavior and saccade dynamics, resulting from minimizing the AED cost.** (A,B) 3D trajectories in the (y,z) and (x,z) views. Although the torsional range is constrained to a width of $\sigma_x$ = 4.88 deg, the eye is not confined to Listing's Plane. (C) Saccade dynamics. Black dots: data from 1500 saccades starting from randomized initial positions in randomized directions. When starting from straight-ahead, saccades follow a clear saturating amplitude-peak velocity relation (red dots: purely horizontal saccades, blue dots: purely vertical saccades; green dots: 45 deg oblique saccades.) Note that vertical saccades are slowest and horizontal saccades are fastest. (D) All trajectories were nearly straight, as most correlations between the horizontal and vertical velocity profiles were near 1.0.

direction-dependence on the saccade dynamics we ran the model once more to elicit saccades from straight-ahead in three selected directions: purely horizontal (red) dots, purely vertical (blue) and oblique (45 deg re. horizontal; green). The dashed curves show the best-fit exponential functions, described by $V_{pk} = V_0(1 - \exp(-\Delta R/\alpha))$.

Note that the horizontal saccades were by far the fastest, while the vertical saccades were much slower. While almost all the torque applied to the eye by the horizontal muscles is converted in horizontal motion, the vertical and oblique muscles divide their torques along the vertical and cyclo-torsional components (see Table 1). Furthermore, because of the pulleys, the

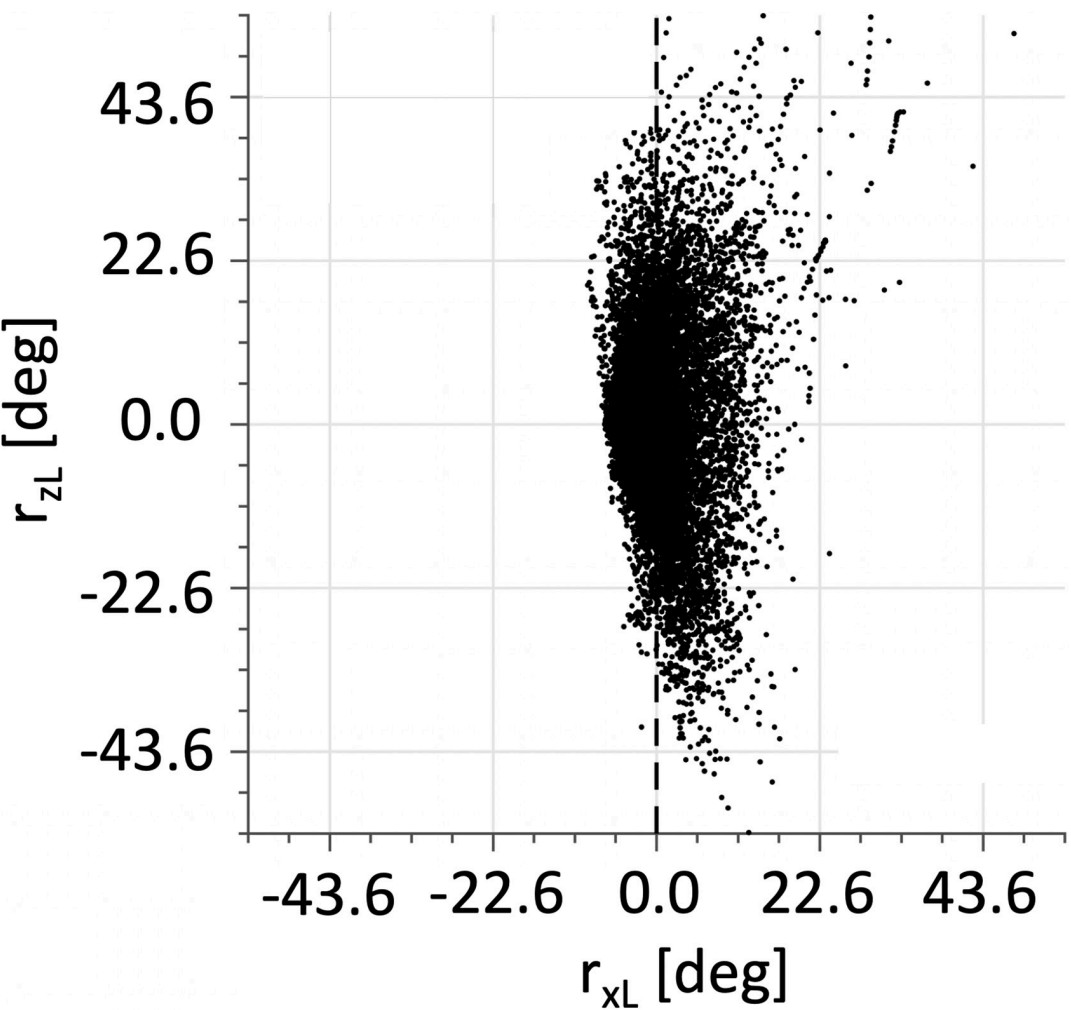

**Fig 10. Same data as in Fig 9, but rotated by $R_p$ (Eq 32), so that it is expressed in the reference frame in which the primary position (the normal to the best-fit plane) coincides with P = [1, 0, 0].** This is defined as Listing's reference frame. Compare to Fig 9B.

force vectors applied to the eye by the vertical and oblique muscles are less orthogonal to the moment arm than the horizontal recti. Together, these factors reduce, on average, the effective torque, hence the peak velocity, in the vertical direction. The velocities of the oblique saccades tended to fall between these two extremes.

In line with the nonlinear main-sequence properties, and the relatively straight saccade trajectories, the model saccades also exhibited strong component cross-coupling. Fig 11 shows how the shorter component of an oblique saccade (here kept fixed at 8 deg in the horizontal direction) is systematically stretched in duration (and lowered in its peak velocity) as the orthogonal vertical component increases in size from 0 to 30 deg (i.e., saccade directions from 0 (horizontal) to nearly 80 deg (upward)). According to the common-source vectorial pulse-generator model of [47], the small-component's peak velocity declines as the cosine of the saccade direction angle with the horizontal meridian (dashed line). The $J_{AED}$ model's result is quite close to this prediction. Note, however, that the common vectorial drive, as proposed by the common-source model, is not implemented in the hardware of our mechanical analogue,

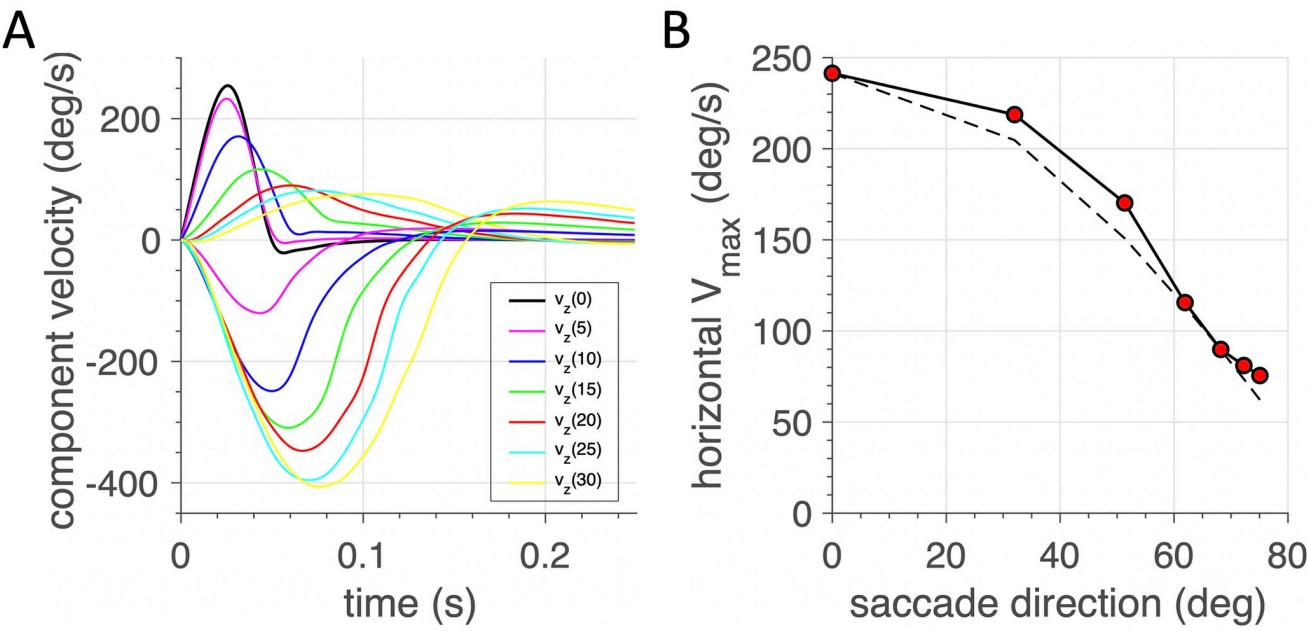

**Fig 11. Component stretching in oblique saccades (after $J_{AED}$ minimization).** In these examples, the horizontal component was kept fixed at 8.0 deg, while the vertical component varied from 0 to 30 deg in 5 deg steps. (A) Horizontal and (inverted) vertical velocity profiles of the oblique saccades. Note that the velocity profiles approximately match in duration. This is especially clear for the larger ($\geq$15 deg) vertical components (green, red, cyan and yellow traces). (B) Peak velocity of the horizontal component depends on the saccade direction. Dashed line: cosine prediction for a common-source vectorial drive ([47] see also Table 3).

as the three driving inputs, $\mathbf{u} = [u_1, u_2, u_1]$ were taken essentially independent. Hence, the observed cross-coupling between the eye-movement components is an emergent property that resulted from the constraints imposed by the optimal control.

## Zero torsion at saccade offset

**3D kinematics and dynamics.**   Although the saccade dynamics resulting from $J_{AED}$ minimization closely follow experimental data, the 3D orientations of the eye still allowed for a wide range of torsional values. Thus, this optimization criterion by itself is not sufficient to explain the emergence of Listing's law. To check whether expanding the accuracy criterion for gaze, which is essentially 2D, to a 3D analogue by requiring that the saccade should have (close to) zero torsion at the end of the movement, we reran the optimization by including the end-point Listing cost in the total cost function: $J_{AEDL_1}$. The result is shown in Fig 12A and 12B. Now, the saccades follow Listing's law, as the standard deviation of all eye orientations around the best-fit plane (which now has its normal vector very close to the $\mathbf{P} = [1, 0, 0]$ direction), has reduced to only 1.7 deg. This value is close to what is observed in monkey and human data [39] (cf. Fig 3).

As can be appreciated, adding the zero-torsion cost did not influence the shape of the saccade-velocity profiles, the straightness of their trajectories, or their main-sequence properties (panel C). There was nearly full component stretching in the oblique saccades (panel D). Thus, the AED minimization by itself forces the eye to behave as a single-axis rotation, and the constraint on the final ocular torsion then suffices to keep the entire eye trajectory near Listing's plane. Including a cost on eye torsion would suggest that the saccadic system has an internal representation of its torsional position state. As argued in the Introduction, experimental evidence seems to supports this idea. For example, any deviation from Listing's plane, even as

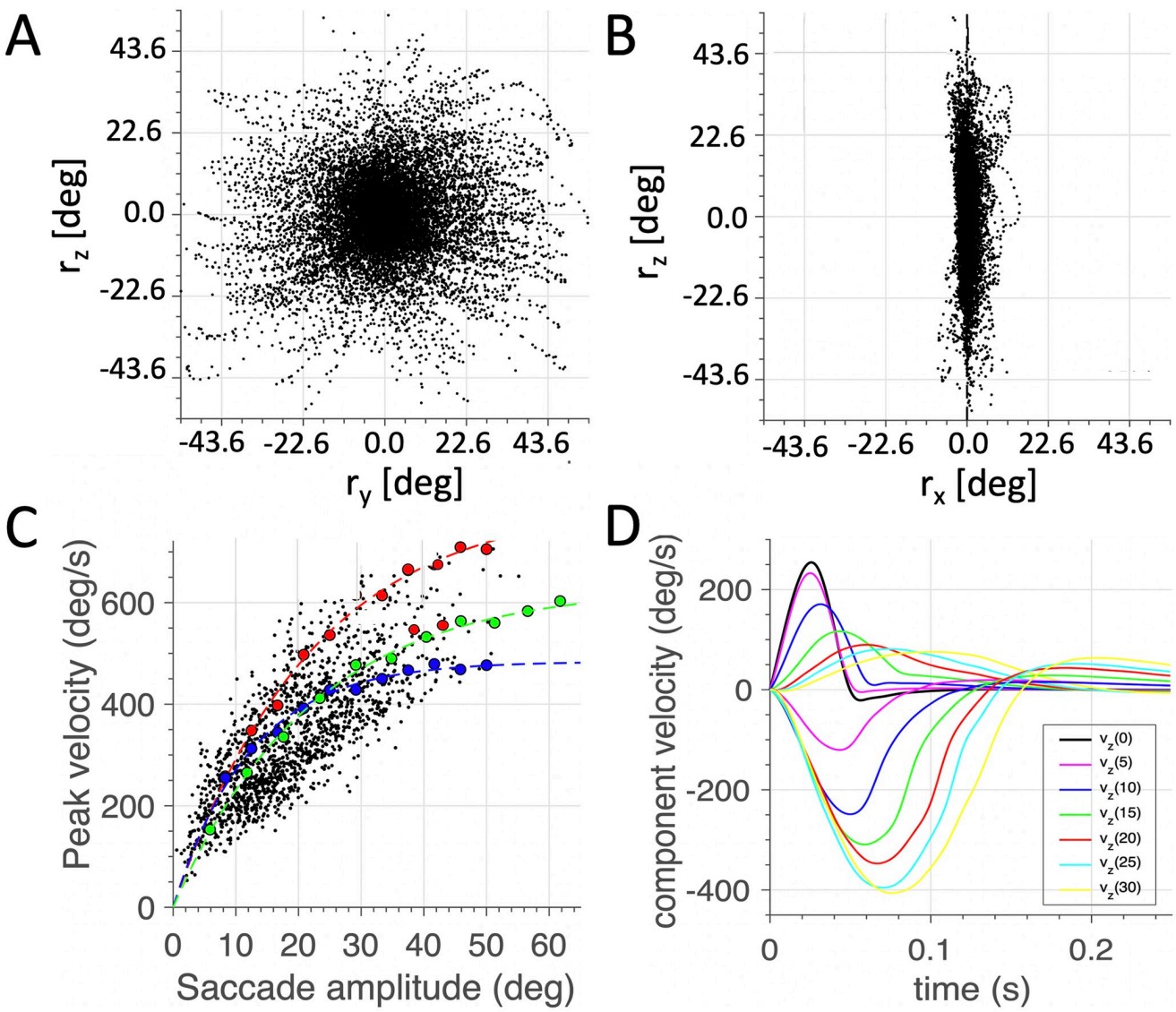

**Fig 12. (A,B) 3D oculomotor behavior and (C,D) saccade dynamics, resulting from minimizing the AED cost and penalizing deviations from Listing's Plane at saccade offset.** Now, the torsional range is constrained to a width of only $\sigma_x = 1.7$ deg (B), which keeps the eye close to Listing's Plane, also during the saccade trajectories. (C) The saccade dynamics still follow the same saturating amplitude-peak velocity relation as in Fig 9C, and nearly straight trajectories (D), as there is clear component stretching.

small as one to two degrees, is typically corrected by the next saccade [18]. Moreover, in more natural 3D gaze-control behaviors, like in eye-head coordination, data suggest that the system appears to plan a gaze trajectory for which the ocular torsion is zero when the entire gaze shift is over [32]. Such a strategy would be in line with a cost that includes eye torsion at the end of the saccade.

## Minimizing fixation-force

**Symmetrical case: $\Delta z = 0$.** Yet, explicitly forcing the eye to have zero torsion at the end of the saccade also presupposes that the orientation of Listing's plane coincides with the straight-ahead direction. This, however, is not supported by experimental data (Fig 3). Thus, we

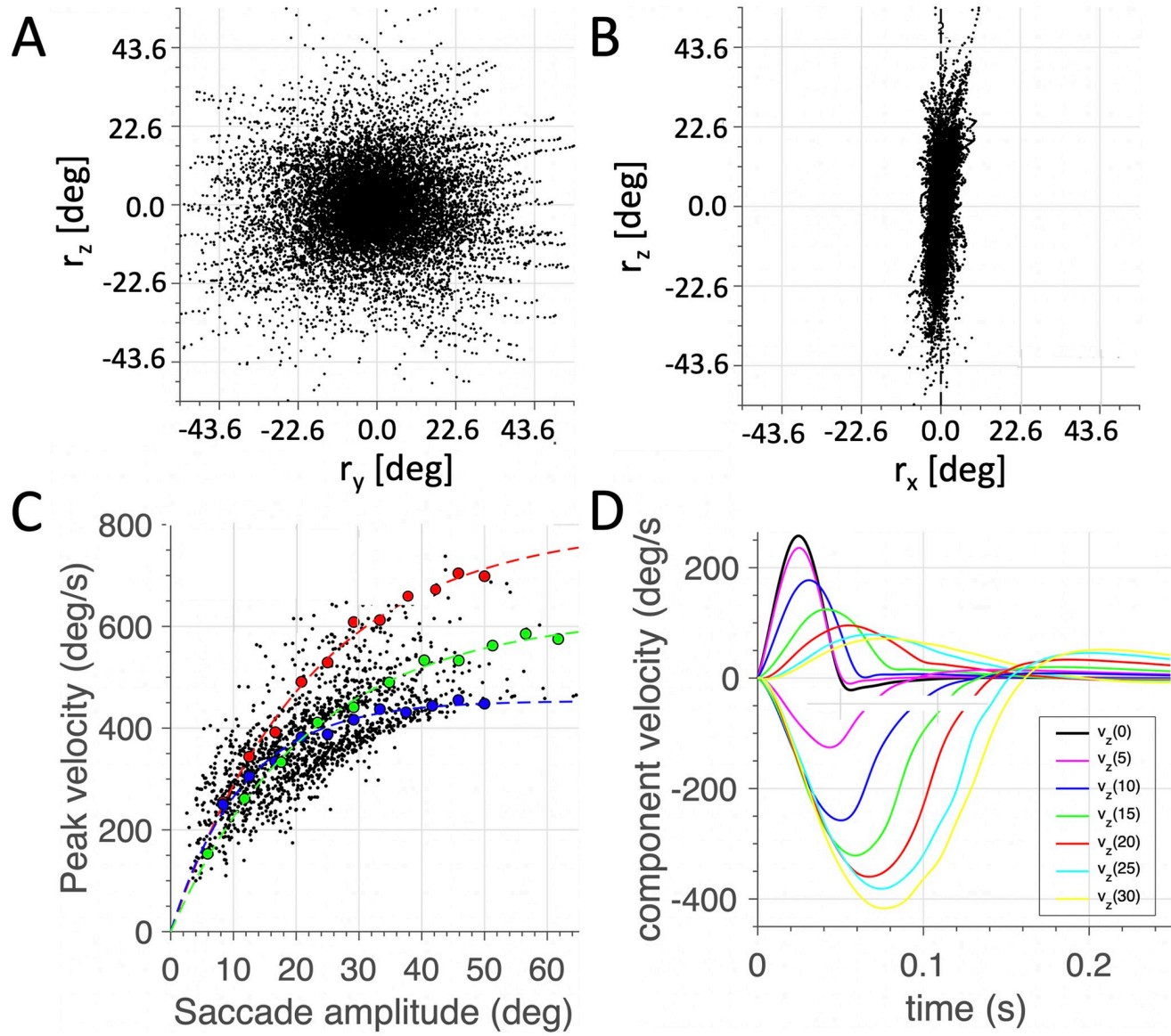

**Fig 13. (A,B) 3D oculomotor behavior and (C,D) saccade dynamics, after minimizing the AED cost and the total force on the eye during fixation.** Default case: $\Delta z = 0$. Now, the torsional range is constrained to a width of only $\sigma_x = 1.25$ deg (B), which is closely in line with Listing's Plane. The saccade dynamics follow a saturating amplitude-peak velocity relation (C). Also, oblique saccades show similar component stretching as in the AED and 3D-target optimizations (D).

wondered how the orientation of Listing's plane would depend on the pulling directions of the extra-ocular muscles. We reasoned that also the forces exerted by the muscles on the eye when it is not moving, are associated with a certain cost. At steady fixation, the brain might want to minimize the total force required to maintain that eye orientation.

In Fig 13 we show the results when including a force cost at fixation for the mechanical prototype shown in Fig 5 and Table 1. Interestingly, the inclusion of force minimization in the optimal control led to good results for all aspects of saccade control. The scattering around Listing's plane resulted to be very small: a width of only 1.55 deg, which is even better than for the $J_{AEDL_1}$ cost functional. The trajectories were approximately straight, with near-complete

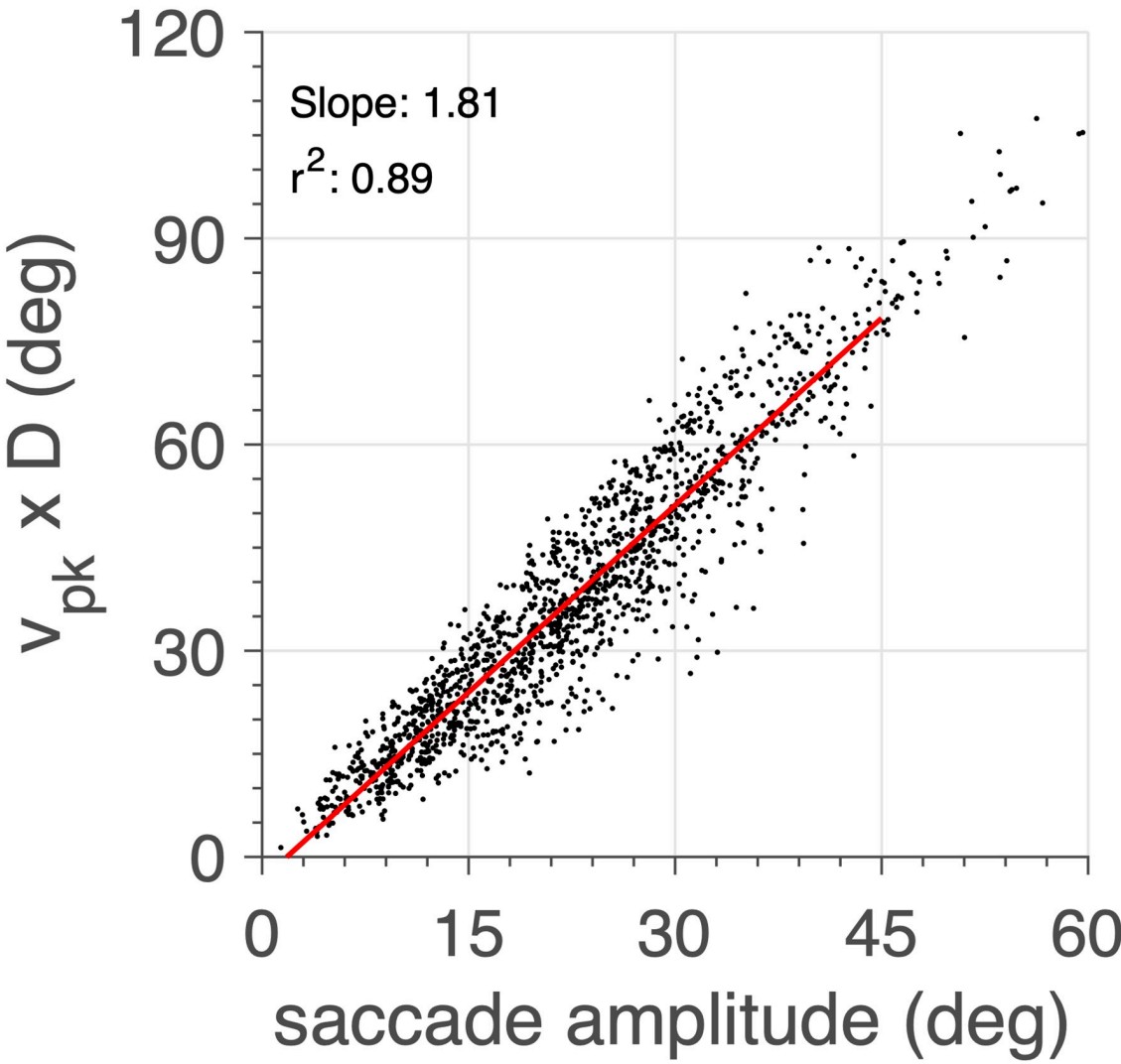

**Fig 14. The model produces a tight relationship ($r^2 = 0.89$) between saccade amplitude and the product between peak eye velocity and saccade duration (quantified at 10 ms resolution) for all saccades shown in Fig 13 (cf. Fig 4C).**

component stretching (panel D), and also the nonlinear properties of the main-sequence were not strongly affected by including the force cost (panel C). The best-fit Listing's plane was pitched by about 4 deg downward. This small downward tilt of the plane probably arose because in the $\Delta z = 0$ case the centers of rotation for the drivers of the vertical recti and obliques were slightly elevated with respect to the eye's horizontal plane (see Table 1; column $P_0^z$).

To further illustrate the appropriate saccade dynamics, Fig 14 shows that for all (slow and fast) saccades, regardless their trajectories and starting points, a tight relation emerged for the saccade amplitude vs. the product of its peak velocity with its duration. The simulated data show a larger variability than is typically observed in the human data (Fig 4C), which is mainly due to the discrete nature of the saccade-duration estimates (here taken at a 10 ms resolution).

**Asymmetrical cases: $\Delta z \neq 0$.** To verify the influence of the muscular geometry on the orientation of Listing's plane, we varied the relative position of the muscles' cranial insertion points with respect to the equatorial plane of the eye. We shifted all insertion points either upwards or downwards with respect to the horizontal ocular midplane, by $\Delta z$, ranging from

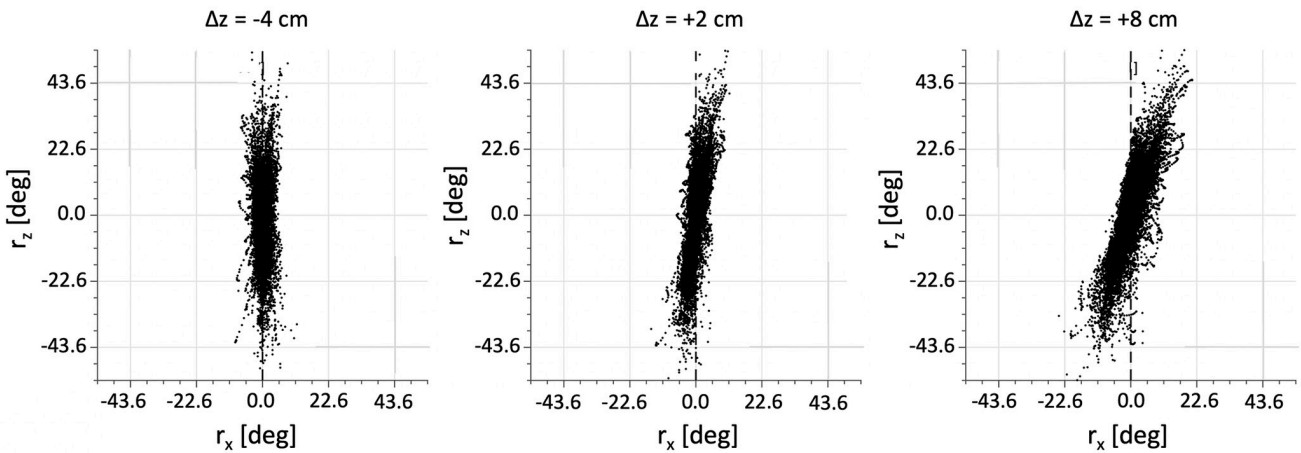

**Fig 15. Eye orientations in the xz-plane for all saccades, for three different vertical shifts, Δz, of the cranial muscle insertion points with respect to the horizontal midplane of the eye.** Note that the orientation of LP tilts as a function of the shift.

-8.0 to +8.0 cm, in nine steps of 2.0 cm. Fig 15 shows the (*xz*) side views of LP for three different shifts: Δz = −4.0, +4.0, +8.0 cm, respectively. Note that the plane pitches downwards in a systematic way with increasing Δz.

Fig 16 shows the empirical nonlinear relationship between the offset Δz and the resulting pitch angle of LP in the *xz*-view. At the largest upward shift, we obtained a downward pitch angle of about -15 deg, which is very similar to what has been obtained in the monkey (Fig 3B).

## Results for all optimizations

Table 3 summarizes the results for all optimization strategies. Although the $J_{AE}$ control strategy (no duration constraint) seems to result in a 3D behavior constrained by Listing's law, it does not generate normal saccade dynamics. Typically, the main sequence resulted to be close to linear, as evidenced by the large angular constant of the saturating exponential curve. The saccade durations for this strategy were all $D = 220$ ms, as no cost was associated to the saccade speed. As a result, saccades were slow, in order to minimize the total energy (proportional to $\|\Delta\mathbf{U}\|^2$) spent by the inputs. Moreover, although saccades were straight, they did not display any component stretching, as the velocity profiles for the horizontal components of oblique saccades did not change with saccade direction ($r_{CS}$ = -0.04, Table 3; see also Supporting Information, S2–S4 Figs). This is to be expected from a linear main sequence: in that case, component stretching is a simple consequence of linear vector decomposition.

Abnormal dynamics were also obtained for the $J_{AD}$ functional (no energy constraint): again the saccade durations were constant, but now at $D = 100$ ms, and the model produced very high saccade speeds (up to 1000 deg/s), as the amount of energy spent by the input signals was not penalized. Component cross-coupling in oblique saccades was observed, but in a completely opposite way as seen in Fig 11: as the vertical component increased in size, the peak velocity of the constant-size horizontal component also increased (rather than decreased) with the saccade direction ($r_{CS}$ = -0.98, Table 3). To keep the final amplitude at 8 deg, however, the horizontal component initially made a large overshoot, and then returned to the final target location during the remaining vertical movement. As a result, the saccade trajectories were

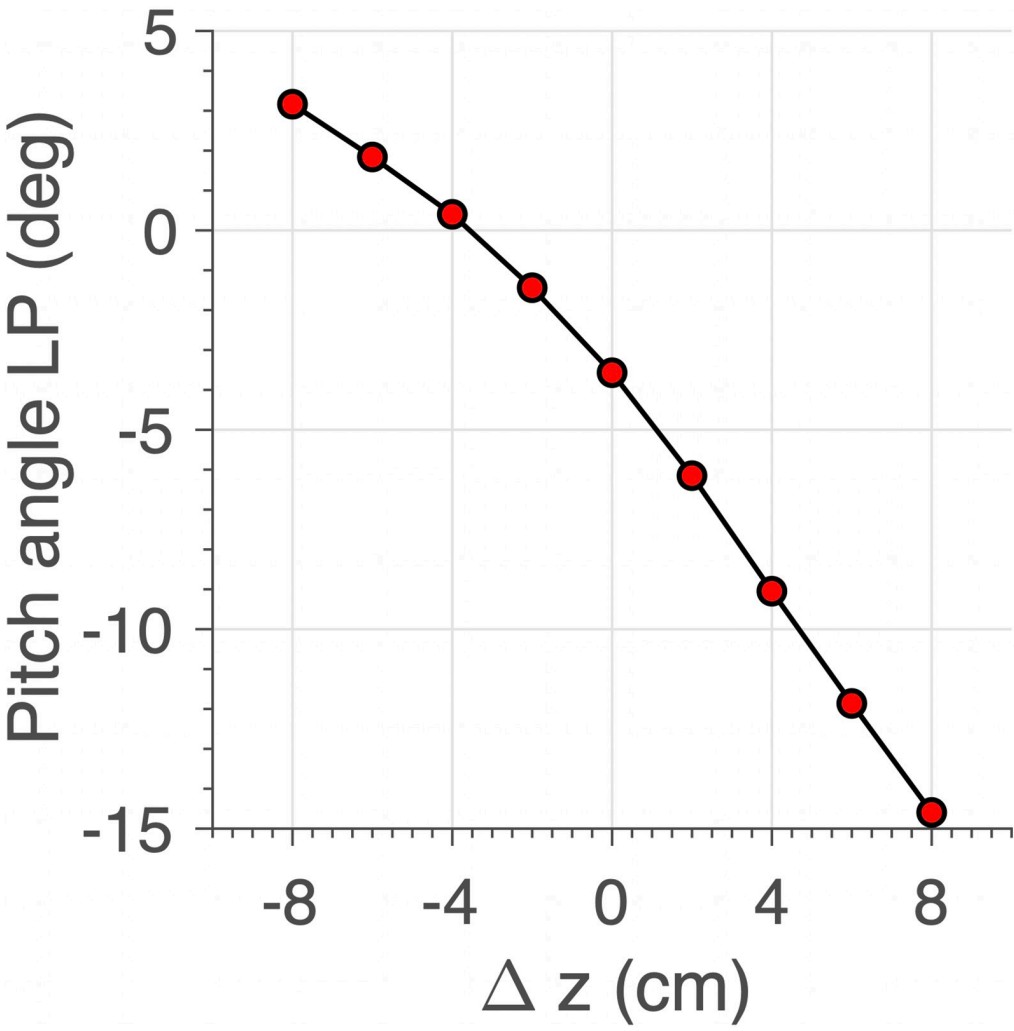

**Fig 16. The pitch angle of Listing's plane varies systematically with the craniocentric insertion points of the muscles re. eye's horizontal plane.** Negative values: downward pitch angle.

strongly curved, and the saccades could not be described as single-axis rotations (see S2–S4 Figs in Supporting Information).

In contrast, imposing torsional constraints on the saccade endpoints, or on the entire saccade trajectories, as well as minimization of the fixation force for all muscle geometries led to both Listing's law, and to normal nonlinear saturating saccade dynamics. In all cases, saccades were nearly straight, and they displayed appropriate component cross-coupling as expected for single-axis rotations.

## Discussion

### Summary

This work accounts for the dynamic and kinematic behaviors of saccadic eye movements and eye orientations in three dimensions, by simulating a simplified mechanical model of the eye,

driven by three independent motor signals, and subjected to different optimal control strategies. Our model explains the major properties of saccade kinematics and dynamics in 3D with a minimum of prior constraints on its biomechanical properties and neural control: direction-dependent nonlinear main-sequence behavior, component cross-coupling, 3D single-axis rotations of saccades, straight trajectories in Listing's plane, and Listing's law.

In our model, all six extraocular muscles were given the same elasticity (and kept constant), the viscous force on the eye was taken proportional to eye angular velocity, and the eye's inertial tensor depended slightly on instantaneous 3D eye orientation. The direction of the torque vector, exerted by each tendon, was determined by the shortest path from the muscle's insertion point on the globe, $\mathbf{Q}_i$, to its craniocentric via point, $\mathbf{X}_i$ (through passive pulleys for the vertical and oblique muscles). Below, we will argue that these simplifications were not essential to obtain the resulting eye-movement properties. Our model provides novel suggestions for the potential control principles underlying the emergence of Listing's Law and the main-sequence in our robotic prototype, and it may serve as a starting point for more realistic biomimetic implementations in humanoid robotic systems.

Although previous studies have applied optimal control principles to explain human (saccadic) oculomotor behavior, they have so far been confined to horizontal saccades only [13, 50–52, 57–59]. These studies suggested that the nonlinear main-sequence behavior of saccades (Fig 4) may result from a neural control strategy that aims to optimize speed-accuracy trade-off, rather than from mere saturation of neuronal peak-firing rates at the level of the brainstem burst generator as proposed by [45]. Indeed, more recent single-unit recordings from the midbrain SC provided supporting evidence for such an optimal control by revealing synchronous firing properties of Superior Colliculus cells in the neural population that encodes the vector of the saccade goal, and its trajectory [48, 49].

## Emerging properties

Our 3D mechanical model accounts for a number of emerging saccade properties, that have so far not been explained by previous models:

i. the nonlinear main-sequence holds for saccades in all directions, and its direction- and initial position-dependence results from the geometric arrangement of the six eye muscles. For example, purely vertical saccades from the straight-ahead initial orientation require a precise coordination of the vertical recti and oblique eye muscles, which have their optimal pulling directions not aligned with the cardinal (vertical and cyclo-torsional) Cartesian axes (see Table 1), and had less-effective, non-orthogonal, moment arms. As a result, vertical saccades of the model were slower than horizontal saccades, which involved an optimal orthogonal activation of the horizontal recti (e.g., Fig 9).

ii. Despite the direction-dependence of the main sequence, the model produced a considerable amount of cross-coupling between the components of the instantaneous eye-velocity vector. This resulted in trajectories in which both velocity components had very similar durations, and were roughly scaled versions of each other (Fig 11A).

iii. Because of this strong component cross-coupling, saccade trajectories were approximately straight in all directions. As a result, the smaller component of the saccade vector was stretched in time, and reached a lower peak velocity than when executed in isolation. This effect could be well described by the cosine (horizontal component, as in Fig 11B) and sine (vertical component; not shown) projections of the vectorial velocity vector, as predicted by the common-source model [47]. This conceptual model assumes that saccades are driven by a central vectorial pulse generator, which represents the vectorial

velocity of the eye, and that the horizontal/vertical/torsional saccade components are jointly derived from this common drive by vector decomposition. As our mechanical model is driven by three essentially independent motors (Fig 5), the apparent common vectorial velocity drive is implicitly encoded by their outputs as a result of the optimal control constraints.

iv. By including the minimization of static total force on the eye, Listing's law emerged without any additional assumptions regarding the eye's cyclo-torsional state, or with the inclusion of precisely positioned pulleys for the horizontal and vertical/torsional muscle pairs. Moreover, this force cost could also account for the orientation of Listing's plane with respect to the head, and hence, for the direction of the primary position with respect to the oculomotor range (Fig 3). Although suggestions in this direction have been made before in the literature [16, 24, 27, 60], our model explicitly demonstrates this potential relationship for the first time.

### Saccade dynamics

The dynamic properties of the saccades require the simultaneous optimization of energy, duration and accuracy ($J_{AED}$ minimization; Fig 8). Putting a constraint on the saccade duration in the form of a hyperbolic discount leads to the saturation of saccadic peak eye velocity for large amplitudes, and the affine relation between saccade duration and amplitude. Adding a penalty on the total energy consumption of the 3D input drive is cause for the strong cross-coupling between the components. This cross-coupling ensures straight saccades (and thereby the shortest path taken by the eye), and can be effectively envisioned to promote a single-axis rotation of the globe from starting to end orientation. Dropping either the penalty on saccade duration ($J_{AE}$) or energy ($J_{AD}$) disrupts the main-sequence and cross-coupling properties of the saccades (Table 3; Supporting Information, S2–S4 Figs). However, it should be noted that the current simulations did not incorporate the effect of signal-dependent (multiplicative) noise in the input driving signals. It can be demonstrated that speed-accuracy trade-off in the presence of signal-dependent noise will show up in a similar quadratic way in the total cost as the energy term in the current formulation (Eq 26), so that the energy term in the cost functional may be dropped when signal-dependent noise is included in the model ([58, 61]; unpublished results from our lab).

### 3D saccade kinematics

The classical cost functional that jointly minimizes the mean accuracy of (2D) target acquisition, input energy expenditure, and movement duration ($J_{AED}$) could not readily account for the 3D behavior of saccades, as the torsional excursions of the model eye resulted to be too large, at about ±10 deg. Thus, to obtain a behavior that would be better in line with Listing's law, the accuracy constraint had to be extended by either explicitly penalizing the amount of ocular cyclo-torsion at saccade offset, or (but perhaps trivially) by constraining the entire trajectory to Listing's plane. Because the AED criterion by itself results in single-axis rotations of the eye, putting an additional constraint on the final eye orientation in 3D would thus ensure that the entire trajectory remains in Listing's plane, by virtue of Eq 12. This extra constraint hardly affected the dynamic behavior of the saccades. The requirement that the entire saccade trajectory should stay in Listing's plane, however, led to much slower saccades than the other cost functionals of Table 2, as the asymptotic velocity was only 440 deg/s. Inclusion of a Listing

cost only at saccade offset generated saccades that could reach velocities of nearly 600 deg/s (Table 3; see also Supporting Information, S5 Fig).

Yet, the explicit requirement of zero cyclo-torsion presupposes that the primary direction is known, and aligned with the straight-ahead direction, i.e., the center of the oculomotor range. Experimental results, however, indicate that the primary direction may be located more eccentric, in the upward viewing direction (see Fig 3, for an example). Although the precise reason for this off-center direction is unclear, our model simulations with the asymmetrical muscle insertions indicate that the geometrical arrangement of the extra-ocular muscles could be directly related to this phenomenon. Interestingly, by adding a static force cost to the overall classical AED cost functional, Listing's law became an emerging property of the system, but now with the primary direction systematically depending on the muscle geometry (Figs 15 and 16). Note that the human eye generates about 3–4 saccades/s, with an average duration of 80–100 ms, which means that 60–70% of the time the eye is not moving, but kept at the eye orientation obtained after the previous saccade. It would therefore make sense for the system to minimize the total effort (i.e., "optimize the skill") needed to keep the eye in any particular orientation.

Our simulations thus suggest that Listing's law (Fig 3) and the main-sequence behavior of saccades (Fig 4) could be part of a joint optimization strategy of the oculomotor system that aims to minimize movement duration and fixation effort when collecting sensory evidence about the environment. In dealing with this problem, the system has to overcome a sluggish (overdamped), anisotropic plant with quite complex static and dynamic mechanical properties [25–27, 35, 38, 62], as well as the noncommutative nature of sequential rotations [3, 6, 16, 19, 37, 39].

Note that Donders' law and Listing's law could also help to optimize visual-sensory functions, like stereoscopic depth perception and the perception of (binocular) visual line orientations seen from different eye positions [21, 33, 34]. The pulley system of the horizontal recti [25, 26] might then also function to facilitate joint binocular control. If so, their plane of action, and even the location of the primary position, could then perhaps reflect an adaptive response of the oculomotor system to the statistics of natural binocular visual inputs [63].

## Model simplifications

Clearly, the mechanical model of Fig 5 is a coarse simplification of the true complexity of the human eye: the actual properties of the muscles were modelled as simple linear springs with a fixed elasticity, taken to be equal for all six tendons (cf., [27, 64]). Moreover, the paths taken by these muscles to exert their forces on the globe were not constrained by (static or dynamic) pulleys [25, 26, 29, 38]. These mechanical simplifications for sure have an influence on the exact motor commands needed to control the plant. This is already demonstrated by the effects of the different geometrical arrangements of the muscles with respect to the eye's equatorial plane (Fig 15), but also by adding head-fixed pulleys for the horizontal recti in our simulator (results not shown). Yet, the major saccade properties: their main-sequence behavior, straight trajectories, component cross-coupling and Listing's law, all emerge for these different geometries, albeit in slightly different ways (e.g., as a tilt of LP). The concept is illustrated in the schematic of Fig 17.

More realistic models of (nonlinear) extraocular eye muscles [27, 64], and incorporation of a pulley mechanism [24–26], will obviously change the model eye's 3D transfer characteristic, but the learning algorithm that uses the eye's measured outputs (3D orientation and angular

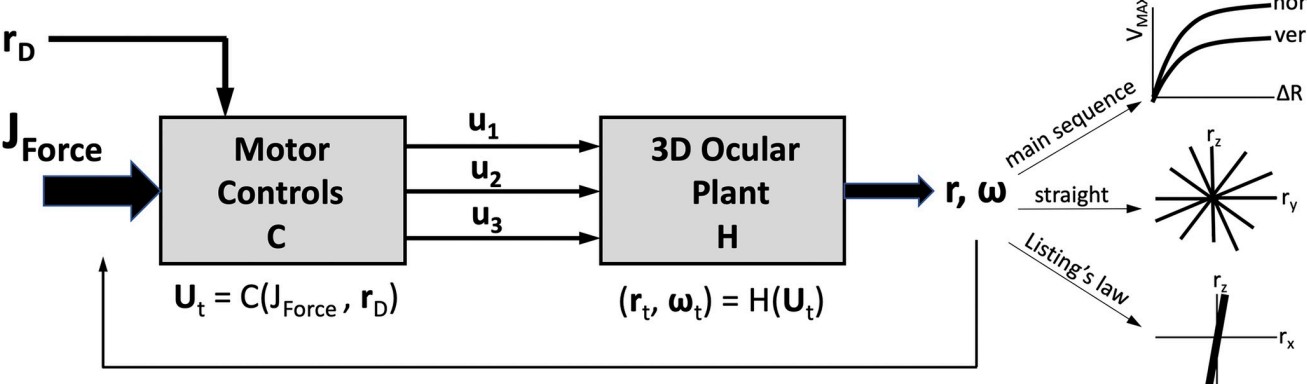

**Fig 17. The total model encompasses two linked (nonlinear) components: the sequence of motor commands, $U_t$, and the 3D eye plant, H.** The plant's output (described by 3D eye orientation, $r_t$, and angular velocity, $\omega_t$) is used to tune the cost function, $J_{Force}(\mathbf{r}, \boldsymbol{\omega})$, that drives the motor outputs. The system's behavior follows the dynamics and kinematics of real saccades (insets, right). Any change in the plant's parameters will require corresponding changes in the motor outputs, $u_t$. $r_D$: 2D target location. The thin feedback arrow represents the training phase during which eye movements recurrently train the motor outputs to minimize the imposed cost.

velocity) in combination with the set of optimization constraints, will result in appropriate (albeit different) commands to cope with more complex plant properties.

## Supporting information

**S1 Fig. Assessing the relative contributions of inertial acceleration and angular velocity to the eye movement (Eq (13)).** (A) Saccade-like traces (generated by an appropriately scaled tanh function): position (in deg) from 2 to 60 deg amplitude; angular velocity (in deg/s); angular acceleration (in deg/s²). (B) Main sequence of the simulated saccades. (C) Instantaneous value of the $I\boldsymbol{\alpha}$ and $\boldsymbol{\omega} \times (I\boldsymbol{\omega})$ terms (taken from t = 0.85–1.15 s). Note differences in scale (factor 100) for the angular velocity (black) and angular acceleration terms. (D) The relative RMS power of the two terms (calculated over the same 300 ms as in C) as function of saccade amplitude. Its value increases with amplitude up to about 0.02% for the largest saccades. As a result, the latter term was ignored in our simulations.
(TIF)

**S2 Fig.** (A,B) 3D oculomotor behavior and (C,D) saccade dynamics, resulting from minimizing the AD cost. Although the 3D eye orientations are close to Listing's law, the saccade dynamics (main sequence and component cross-coupling) are abnormal. The saccade peak velocity (C) doesn't seem to saturate, as the exponential fit yields a non-physiological asymptote near $10^4$ deg/s, with a angular constant close to the oculomotor range. (D) In the fixed-component oblique saccade test (as in Fig 9), the 8 deg horizontal component develops a large overshoot (with all durations the same, at about 100 ms), which is followed by a long return phase. This results in goal-directed, but curved saccade trajectories. See also Table 3 for numerical details.
(TIF)

**S3 Fig. (A,B) 3D oculomotor behavior and (C,D) saccade dynamics, resulting from minimizing the AE cost.** Also for this control strategy the saccades are abnormal: (C) the peak velocity increases linearly with saccade amplitude (but saccades remain much slower than for the $J_{AD}$ strategy, with the fitted asymptote at around 650 deg/s), and absence of component stretching (D), as all horizontal saccade components have the same, slow, velocity profile (all

positive traces superimpose with a fixed duration of about 200 ms; See also Table 3 for numerical details).
(TIF)

**S4 Fig. Amount of stretching of the horizontal peak eye velocity component (kept fixed at an amplitude of 8 deg, for saccade vectors in different directions, as indicated on the abscissa; see also panels D in S2 and S3 Figs) for the AD (A) And AE (B) cost-minimization strategies.** The simulated data have either a negative, or no correlation with the expected common-source cosine prediction (dashed lines; cf. Fig 11), indicating that saccade trajectories are curved. See also Table 3.
(TIF)

**S5 Fig. A,B) 3D oculomotor behavior and (C,D) saccade dynamics, resulting from minimizing the $J_{AEDL_2}$ cost.** As expected, saccades follow Listing's law (width of the plane is 2.25 deg), but peak velocities (C) of the saccades are much lower than for the strategies constraining AED and AED with final eye position in LP. (D) Saccade trajectories are straight, as the correlations between horizontal and vertical velocity profiles are high (see also Table 3 for numerical details).
(TIF)

## Acknowledgments

We thank prof. José Santos-Victor of the Computer and Robot Visual Laboratory of the Institute Superior Técnico in Lisbon,Portugal, for valuable feedback.

## Author Contributions

**Conceptualization:** Akhil John, Carlos Aleluia.

**Data curation:** Akhil John, A. John Van Opstal.

**Formal analysis:** A. John Van Opstal, Alexandre Bernardino.

**Funding acquisition:** A. John Van Opstal.

**Investigation:** Akhil John, Carlos Aleluia.

**Methodology:** Akhil John, Carlos Aleluia.

**Project administration:** A. John Van Opstal, Alexandre Bernardino.

**Resources:** Akhil John, Alexandre Bernardino.

**Software:** Carlos Aleluia.

**Supervision:** A. John Van Opstal, Alexandre Bernardino.

**Validation:** Akhil John.

**Visualization:** Akhil John, Carlos Aleluia.

**Writing – original draft:** Carlos Aleluia, A. John Van Opstal.

**Writing – review & editing:** Akhil John, A. John Van Opstal, Alexandre Bernardino.

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
