## [Decision Letter · Decision Letter 0]

9 Dec 2020

Dear Prof. Bernardino,

Thank you very much for submitting your manuscript "Modelling 3D Saccade Generation by Feedforward Optimal Control" for consideration at PLOS Computational Biology.

As with all papers reviewed by the journal, your manuscript was reviewed by members of the editorial board and by several independent reviewers. In light of the reviews (below this email), we would like to invite the resubmission of a significantly-revised version that takes into account the reviewers' comments.

All reviewer noted that this was an interesting and important study. However, they also all had concerns regarding clarity and omission of certain aspects in the model. While this will require significant work - mostly in terms or re-writing - I believe that this is doable and will significantly benefit the manuscript.

We cannot make any decision about publication until we have seen the revised manuscript and your response to the reviewers' comments. Your revised manuscript is also likely to be sent to reviewers for further evaluation.

Sincerely,

Gunnar Blohm, Ph.D.

Associate Editor

PLOS Computational Biology

Wolfgang Einhäuser

Deputy Editor

PLOS Computational Biology

Reviewer's Responses to Questions

**Comments to the Authors:**

Reviewer #1: While this is an interesting paper, I had to give it two full reads to understand where the authors were going with the paper. Ultimately, I reached the impression that it was written by two teams of authors, one with substantial physiological experience in the primate oculomotor system, and another with substantial experience in robotics and modeling. There is certainly nothing wrong with such a hybrid approach, but the paper seems more a concatenation of these two conceptual realms rather than a seamless integration. It is recommended that the authors revise the paper with attention to making it more readable, and one way to do that would be to better define the goal of the paper at the outset.

Overall, the authors do support the case that penalties for accuracy, energy, and duration are insufficient to generate Listing's Law and hence commutativity of ocular rotations, but either the imposition of a penalty for the violation of Listing's Law at the saccade end point, or of total muscle force, give rise to reasonable Listing's type torsional behavior. This, of course, is applicable to the simplified robotic eye simulated here, but it would be nice if the authors made it clear from the outset that they are dealing with such a simplified robotic eye, rather than attempting to make it very representative of the actual biologic oculomotor system. The omission of explicit pulleys, of course, substantially alters the properties of the oculomotor actuator, and thus would leave somewhat confusing the relationship between Listing's Law as commanded entirely neurally, versus implemented to a large extent by the orbital connective tissue pulley system as appears to be the biologic case. Certainly, the controversy over this issue from about two decades ago was largely settled after the work of Ghasia et al. and Kleier et al. from the Angelaki Laboratory. The actual biological system apparently does not implement Listing's Law by commands to the cyclovertical muscles for torsion, although certainly during eye-head coordination and vestibulo-ocular reflexes there exist violations. While this difference between the robotic eye modeled here and the biological one does not defeat the purpose of the paper, it would be worth making it clearer at the outset to the reader that there is a difference here that may be important.

Figure 1, and throughout the paper. This reviewer finds it confusing and annoying that the authors seem to slip seamlessly between representation of angles as radians versus degrees, even within the same figure and its legend. It would be very helpful if the authors would consistently employ one system or the other, but not both. I prefer degrees, as does much of th published oculomotor physiology literature.

Lines 42 through 46. Here the authors make some oblique references to the existence of the pulley system in the actual biological eye plant. Of course, there has been much development of the pulley idea since 1995, and the authors really should be citing the more recent references to the behavior of the peripheral oculomotor connective tissue apparatus and the muscle layers.

Lines 48 and 49. The authors seem to be implying that Tenon described the pulleys in 2003, but in fact that is a reference to a translation of Tenon's work in 1816. Moreover, Tenon did not have much to say about the kinematic behavior that pulleys might have, so this reference is not too helpful to the reader. Tenon’s descriptions were barely even qualitative concepts of the anatomy.

Page 3, lines 59 through 86. There is an extensive numerically labeled series of paragraphs about Listing's Law. While correct, much of this seems irrelevant to the message of the current paper, and probably should be deleted in the interest of brevity.

Lines 90 through 95. Of course, the empirical observations cannot be accounted for by a passive pulley mechanism. Although the authors here mention the existence of active pulleys, it would be useful to say a bit more about their behavior, or else emphasize that the current model is not intended to have very much biological realism. Actual biological pulleys are active.

Lines 138 through 143. Here, the authors reveal clearly for the first time that they intend to analyze a simplified robotic eye rather than the actual biologic system. That is fine for theoretical purposes, but the robotic eye needs to be described in much greater detail than in Figure 3B. The model does not have any pulleys in the biological sense, and is insufficiently described. Based on the use of centimeter dimensions later in the paper, one gets the impression that the authors may be talking about a large ophthalmotrope that might have dimensions tenfold greater than that of the actual human eye and oculomotor system. The authors should greatly expand their description of the model eye and clarify that it is not intended to be particularly biological.

Line 163. The equation here including the imaginary numbers would seem to include some errors, because it is hard for this reviewer to understand how the product of ijk could be -1.

Lines 190 and 191. This sentence does not on its face have much relevance to the paper and probably should be deleted as it is distracting.

Lines 210 through 221. Here the authors do explicitly disclose that their model does not replicate actual anatomic features of muscles. They also disclose that the way muscles change their direction of action and eye orientation changes is not physiologic either. That disclosure should be made earlier in the paper.

Table #1. The coordinates are specified in centimeters, rather than physiologic millimeters. This is okay for modeling purposes, but it took me as a reader some time to begin to understand that no attempt is being made here to make the model dimensionally realistic.

Lines 229 and 230, equation 18. It appears there may be an error in this equation, since the second equal sign probably is meant to be a plus sign.

Line 241. Which term is the right-handed term of the equation that was neglected in the simulator? It would appear that there are multiple such terms, and with two equal signs in the equation it is hard for the reader to interpret what is stated.

Line 270. How do the authors justify dynamic frictional force at this value? Assumed?

Line 295. Please define the acronym "PRBS."

Page 18, Figure 8. Although in the text the meaning of the green dot is defined, it should be defined in the legend. In the lower right pane, the ordinate scale should be defined, although it is presumably the number of saccade events. It would be helpful to label the panels with the ABCD notation.

Figure 9. It would be helpful if the legend to this figure were a little bit more descriptive. It is not evident from the flow of the text what the message is or why this figure was included. That could be clarified.

Lines 491 and 492. The authors argue that the relative slowness of vertical saccades is because of mechanical actions between the obliques and vertical rectus muscles. This may be so, but it is not persuasive to merely state it without providing some evidence. It would be useful here and perhaps at other points in the paper to specify what the actual mechanical states are of the actuators.

Figure 14. The legends to most of the labels in the figure are too small to be legible. These appear to be smaller than 4-point font, and I could not read them at all when I printed out the paper.

Figure 15 legend. Once again, the centimeter units here are confusing, although I now suspect that the robotic model is much larger than the actual oculomotor apparatus that exists biologically.

Line 606. The paper correctly describes the model as having three independent motor signals for the three antagonist muscle pairs. It would be useful if this was more clearly stated at the beginning that this is not the way things work biologically, although that does not detract from the ultimate conclusions. Similarly, line 611 suggests that the muscles follow the shortest paths from their origins to insertions on the globe, a statement also inconsistent with what has been learned about the actual muscle paths through orbital imaging. Therefore, the conclusion in lines 613 through 615 that the model provides novel suggestions for potential principles underlying the neurobiological oculomotor system is rendered at least somewhat dubious. On lines 615 and 616, it is certainly plausible that this has direct applicability to robotic systems.

Lines 636 and 637. This reader is still either unconvinced or does not understand the argument for the relative slowness of the vertical saccades. That idea deserves to be developed better.

Lines 654 and 655. It is an intriguing and important conclusion that force minimization can give rise to Listing's Law without other assumptions about the cyclotorsional state. That is a useful principle, although the behavior of the connective tissue apparatus and pulley system in the biological eye provides at least another basis for understanding the existence of Listing's Law.

Lines 754 through 760. It does not appear necessary for this paper to argue the alignment of vestibular canals in the extraocular muscles. This alignment is transient and approximate, and as the work of Ben Crane et al. has shown depends on the history of prior vestibular experience of the subject. This section should probably be deleted since the current model does not have any way of accounting for the biologic behavior.

Lines 770 through 772. The authors should probably not stretch the point so much about the optimal control strategy and its resemblance to the human and oculomotor systems without providing data on the actual commands supplied to the muscles. In the absence of that comparison, the similarity is not convincing at the level of fine detail.

Supplemental Figure 1. It would be a favor to the reader to consistently use degrees or radians to represent angles. The reference for panels C and D talks about "the two terms," but leave the nature of these terms unclear.

Supplemental Figure 2. The legend to this figure is insufficiently detailed. Presumably, the authors expect the readers to compare this figure with some of the other named figures in the text, but the legend to this supplemental figure should really stand on its own.

Supplemental Figure 4. The symbols and dotted lines are insufficiently described. The legend to this figure should stand on its own.

Supplemental Figure 5. Once again, the legend is inadequate. The legend should stand on its own.

Reviewer #2: Review is uploaded as an attachment (docx)

Reviewer #3: The paper is about modelling eye movement dynamics using optimal control, specifically addressing the saccade generation. The paper reads like a very good survey of the field, yet perhaps misses out a few optimal control related papers that I am aware of. See for example:

[1] Kardamakis and Moschovakis, Optimal control of gaze shifts.

[2] Saglam, Glasauer and Lehmen, Vestibular and Cerebellar contribution to gaze optimality.

Using optimal control strategy on a model of the eye, the paper tries to explain Listing's Law, straightness of the rotation vector trajectory on the Listing's plane and a main-sequence non linearity that has to do with acceleration and deceleration of the eye as a function of the amplitude of the saccade.

I would have liked that the paper explained Fig. 3(B), the mechanical model of the eye better. Likewise, Fig. 4, could be explained better, as well. The line diagrams are really not clear. Is there a `easily inderstandable' connection between the model and the human eye muscles?

Overall the paper claims that the model explains various properties of the saccade by choosing various different type of optimality criterion. A slight tilt of the Listing's plane is connected to the static force on the eye.

The paper could be considerably shortened. The focus of the paper should be on the model, that explains various properties of a saccade generator and how the components of the optimality criterion is connected to these properties. Connecting to robotic or humanoid control was not necessary and is not done well. At the end, the paper reads like a long winded prose that does not seem to end.

**Have all data underlying the figures and results presented in the manuscript been provided?**

Reviewer #1: Yes

Reviewer #2: Yes

Reviewer #3: Yes

PLOS authors have the option to publish the peer review history of their article (what does this mean?). If published, this will include your full peer review and any attached files.

Reviewer #1: No

Reviewer #2: **Yes: **T. Scott Murdison

Reviewer #3: No
---

## [Decision Letter · Decision Letter 1]

18 Apr 2021

Dear Prof. Bernardino,

We are pleased to inform you that your manuscript 'Modelling 3D Saccade Generation by Feedforward Optimal Control' has been provisionally accepted for publication in PLOS Computational Biology.

Best regards,

Gunnar Blohm, Ph.D.

Associate Editor

PLOS Computational Biology

Wolfgang Einhäuser

Deputy Editor

PLOS Computational Biology

Reviewer's Responses to Questions

**Comments to the Authors:**

Reviewer #2: I appreciate the efforts made by the authors to fully answer my questions and concerns, and to improve the clarity of the manuscript. I think this makes a nice contribution to the field!

**Have the authors made all data and (if applicable) computational code underlying the findings in their manuscript fully available?**

Reviewer #2: None

PLOS authors have the option to publish the peer review history of their article (what does this mean?). If published, this will include your full peer review and any attached files.

Reviewer #2: No

---

## [Editor Report · Acceptance letter]

6 May 2021

PCOMPBIOL-D-20-01801R1 

Modelling 3D Saccade Generation by Feedforward Optimal Control

Dear Dr Bernardino,

I am pleased to inform you that your manuscript has been formally accepted for publication in PLOS Computational Biology. Your manuscript is now with our production department and you will be notified of the publication date in due course.

With kind regards,

Katalin Szabo
